# A deeply conserved protease, acylamino acid-releasing enzyme (AARE), acts in ageing in Physcomitrella and Arabidopsis

Sebastian N. W. Hoernstein [1], Buğra Özdemir[1,5], Nico van Gessel [1], Alessandra A. Miniera[1], Bruno Rogalla von Bieberstein [1,6], Lars Nilges[1], Joana Schweikert Farinha [1,7], Ramona Komoll[1,8], Stella Glauz[1], Tim Weckerle[1,9], Friedrich Scherzinger [1,10], Marta Rodriguez-Franco [2], Stefanie J. Müller-Schüssele[3] & Ralf Reski [1,4✉]

Reactive oxygen species (ROS) are constant by-products of aerobic life. In excess, ROS lead to cytotoxic protein aggregates, which are a hallmark of ageing in animals and linked to age-related pathologies in humans. Acylamino acid-releasing enzymes (AARE) are bifunctional serine proteases, acting on oxidized proteins. AARE are found in all domains of life, albeit under different names, such as acylpeptide hydrolase (APEH/ACPH), acylaminoacyl peptidase (AAP), or oxidized protein hydrolase (OPH). In humans, AARE malfunction is associated with age-related pathologies, while their function in plants is less clear. Here, we provide a detailed analysis of *AARE* genes in the plant lineage and an in-depth analysis of AARE localization and function in the moss Physcomitrella and the angiosperm Arabidopsis. *AARE* loss-of-function mutants have not been described for any organism so far. We generated and analysed such mutants and describe a connection between AARE function, aggregation of oxidized proteins and plant ageing, including accelerated developmental progression and reduced life span. Our findings complement similar findings in animals and humans, and suggest a unified concept of ageing may exist in different life forms.

[1] Plant Biotechnology, Faculty of Biology, University of Freiburg, Schaenzlestrasse 1, 79104 Freiburg, Germany. [2] Cell Biology, Faculty of Biology, University of Freiburg, Schaenzlestrasse 1, 79104 Freiburg, Germany. [3] Molecular Botany, Department of Biology, Technical University of Kaiserslautern, Erwin-Schrödinger-Strasse 70, 67663 Kaiserslautern, Germany. [4] Signalling Research Centres BIOSS and CIBSS, Schaenzlestrasse 18, 79104 Freiburg, Germany. [5] Present address: Euro-BioImaging Bio-Hub, EMBL Heidelberg, Meyerhofstraße 1, 69117 Heidelberg, Germany. [6] Present address: Department of Anesthesiology, University Hospital Würzburg, Oberduerrbacher Strasse 6, 97072 Würzburg, Germany. [7] Present address: Institute for Molecular Biosciences, Goethe University Frankfurt, Max-von-Laue-Str. 9, 60438 Frankfurt, Germany. [8] Present address: Heraeus Medical GmbH, Philipp-Reis-Straße 8-13, 61273 Wehrheim, Germany. [9] Present address: Zymo Research Europe GmbH, Muelhauser Strasse 9, 79110 Freiburg, Germany. [10] Present address: Centre for Integrative Biodiversity Research (iDiv) Halle-Jena-Leipzig, Puschstrasse 4, 04103 Leipzig, Germany. ✉email: ralf.reski@biologie.uni-freiburg.de

Reactive oxygen species (ROS) are by-products of $O_2$ metabolism and represent a challenge to all aerobic life. ROS play a dual role as potentially lethal oxidants and as signaling molecules[1]. Therefore, aerobic organisms possess sophisticated redox systems to scavenge and detoxify excess ROS. The major sources of ROS in plant cells are the electron transport chains in mitochondria and plastids, but also in peroxisomes and at the plasma membrane[2]. Environmental stresses such as heat, drought, or intense light are factors for increasing ROS production to detrimental levels. Plants possess a repertoire of detoxifying enzymes such as catalases, superoxide dismutases, ascorbate peroxidases, glutathione peroxidase-like proteins, and peroxiredoxins. Electrons for reduction are largely provided via non-enzymatic components such as ascorbic acid, glutathione, and NADPH[3–7]. In addition, a range of heat-shock proteins assist in disaggregation or refolding of damaged proteins[8–10].

Despite conversion into non-toxic derivatives, the continuous exposure to ROS results in oxidation of DNA, lipids, and proteins[5]. On the protein level, ROS lead to irreversible cysteine oxidation, advanced glycation end products, adducts with ROS-generated reactive aldehydes, and carbonylation of amino acid side-chains[11,12]. If not cleared via proteolysis, an excess of oxidized proteins accumulates to cytotoxic protein aggregates. Plant antioxidant systems and the role of ROS as signaling molecules in abiotic stress responses are well studied[13]. Yet, factors involved in a plant cell's last line of defense, such as the proteolytic systems for the clearance of irreversibly oxidized proteins, are still underexplored.

A class of serine proteases is evolutionary deeply conserved as their activity is found in bacteria, archaea, animals and plants, and can degrade irreversibly oxidized proteins[14–17]. These proteases have different names in different organisms, e.g. acylamino acid-releasing enzyme (AARE), acylpeptide hydrolase (APEH/ACPH), acylaminoacyl peptidase (AAP), or oxidized protein hydrolase (OPH)[18] but are collectively addressed as AARE here. AARE acts in multimeric complexes[14,17] as a bifunctional protease as it cleaves $N^\alpha$-acetylated amino acids from oligopeptides via an exopeptidase mode, but also cleaves oxidized proteins via an endopeptidase mode[16,19–22]. AARE isoforms from various organisms show different specificities towards $N^\alpha$-acetylated amino acids, with bacterial and archaeal enzymes preferring AcLeu and AcPhe substrates[15,23–25] and plant and animal isoforms preferring AcAla, AcMet or AcGly substrates[16,17,26,27]. Besides substrate specificities, their subcellular localization appears to be conserved among eukaryotes, as human (HsACPH) and Arabidopsis (AtAARE) AAREs are reported as cytosolic enzymes[20,21].

In humans, AARE malfunction is linked to different types of cancer[26,28,29] and sarcoma cell viability[30]. Moreover, AARE and proteasomal activity correlate and cooperatively prevent cytotoxic aggregate formation[31–33]. Several selective inhibitors have been identified[34,35] and blocking of AARE function is considered as anti-cancer treatment[28]. Despite an increasing number of studies on AARE functionality in humans, AAREs in plants are far less characterized.

AARE from Arabidopsis thaliana (AtAARE) and from cucumber have endo- and exopeptidase functions[17], and silencing of AtAARE increased the levels of oxidized proteins[21]. AtAARE activity was also detected in plastid stroma fractions, although a fusion with a fluorescent reporter did not co-localize with chloroplasts[21]. Suppression of AtAARE via RNAi resulted in an enhanced accumulation of oxidized proteins in roots and enhanced electrolyte leakage in leaves, but a further impact on plant physiology was not described[21].

Moreover, the complete loss of function of this protease has not yet been reported for any organism, neither bacterium,

archaeon, animal, or plant. Thus, although the deep evolutionary conservation of AARE suggests its pivotal role in all major life forms, experimental evidence is far from optimal.

In a proteomics study on protein arginylation we identified an AARE homolog from the moss Physcomitrella[36]. Our further analysis revealed altogether three Physcomitrella AARE homologs (PpAARE1-3). Here, we analyzed the subcellular localization of these PpAARE isoforms and of their homolog from the angiosperm Arabidopsis (AtAARE). We show that an alternative splicing event is targeting PpAARE1 to chloroplasts, mitochondria, and the cytosol. We provide evidence that an alternative translation initiation is sufficient to localize AtAARE to the same three subcellular compartments. Bioinformatic analyses of several genomes suggest that the localization of AARE in chloroplasts and mitochondria is conserved across the plant lineage. Employing combinatorial gene knockouts and protein co-immunoprecipitation we found distinct interactions between these three isoforms and their concerted action on progressive ageing in Physcomitrella. Likewise, an Arabidopsis AARE loss-of-function mutant exhibits enhanced levels of oxidized proteins and accelerated bolting, as a hallmark of plant ageing.

## Results

**AARE gene family expansion and splice variants.** Previously, PpAARE1 (Pp1s619_3V6.1) was identified as the most prominent target for N-terminal arginylation in Physcomitrella[36,37]. N-terminal arginylation mediates poly-ubiquitination via the N-degron pathway, thus triggering subsequent proteasomal degradation[38]. Simultaneously, two homologs (PpAARE2: Pp1s108_134V6.1 and PpAARE3: Pp1s97_68V6.1) were identified, although those were not proven arginylation targets[36]. Meanwhile, a new Physcomitrella genome version with chromosome assembly and updated gene versions was released[39]. Consequently, the gene accessions used here are PpAARE1 (Pp3c2_30720V3.1), PpAARE2 (Pp3c12_21080V3.1), and PpAARE3 (Pp3c7_25140V3.1). According to OrthoMCL[40,41], all three proteins are homologs of the Arabidopsis thaliana acylamino acid-releasing enzyme (AtAARE: AT4G14570). According to publicly available data (https://peatmoss.online.uni-marburg.de)[42] PpAARE1-3 are expressed in all major Physcomitrella tissues and developmental stages, although at varying levels (Fig. S1a). Except for leaves (phylloids) and spores, PpAARE1 is the most strongly expressed gene of this family (between 4 and 20 times). In contrast, PpAARE2 is expressed considerably stronger than PpAARE1 and PpAARE3 in spores (Fig. S1a). Likewise, AtAARE is expressed in all major Arabidopsis tissues (Fig. S1b, data utilized from Mergner et al.[43] and downloaded from http://athena.proteomics.wzw.tum.de/). These data indicate a strong, positive correlation between transcript level and protein abundance across all tissues (Fig. S1b). Stress conditions decrease AARE expression in Arabidopsis shoots and in Physcomitrella protonemata (Fig. S1c–e).

To investigate whether other plants also possess multiple AARE homologs and to infer their phylogenetic relation, we conducted BLASTP[44] searches against protein models from selected species using the protein sequence of AtAARE as a query. We selected the alga Chlamydomonas reinhardtii[45], the liverwort Marchantia polymorpha[46], the peat moss Sphagnum fallax (Sphagnum fallax v1.1, DOE-JGI, http://phytozome.jgi.doe.gov/), the lycophyte Selaginella moellendorffii[47], the monocot Oryza sativa[48] and the dicot Populus trichocarpa[49], all available at the Phytozome12 database (https://phytozome.jgi.doe.gov). Additionally, we performed an NCBI BLASTP search against the charophyte Klebsormidium nitens proteome[50], and identified a single homolog (GAQ80280.1) in this species. We also included proteins of Funaria hygrometrica[51], a close relative to Physcomitrella from the Funariaceae family[52], in our search. Finally, the

AtAARE sequence was again searched against the Physcomitrella[39] and Arabidopsis[53] proteomes. Homology of the resulting *BLAST* hits was confirmed if the reciprocal best *BLAST* hit against *A. thaliana* was again AtAARE.

In *P. trichocarpa*, we identified a single homolog for which three distinct splice variants are annotated (Potri.008G160400.1, Potri.008G160400.2, Potri.008G160400.3). These encode different protein isoforms, but two variants seem to encode non-catalytic proteins. AARE enzymes are prolyl-oligopeptidases with a conserved catalytic triad (Ser/Asp/His) in the C-terminal peptidase domain[54,55]. In splice variant 2 (Fig. S2, Potri.008G160400.2) alternative splicing results in the deletion of the catalytic Asp whereas the whole catalytic triad is lacking in splice variant 3 (Potri.008G160400.3). Hence, we consider these splice variants as non-active and disregard them from further discussion.

In rice, we identified two homologs (LOC_Os10g28020.3, LOC_Os10g28030.1), with an additional splice variant (LOC_Os10g28020.1) at one locus which encodes an N-terminal extension.

In *C. reinhardtii*, *M. polymorpha*, *S. fallax* and *S. moellendorffii*, we identified a single ortholog each. In *M. polymorpha*, three distinct splice variants are annotated (Mapoly0111s0036.1, Mapoly0111s0036.2, Mapoly0111s0036.3). The latter two are UTR (untranslated region) splice variants, thus resulting in the same protein sequence, whereas Mapoly0111s0036.1 encodes an N-terminal extension of 97 aa compared to the other two variants. In *F. hygrometrica* we identified three distinct isoforms.

Finally, our *BLASTP* searches using the latest Physcomitrella protein models[39] confirmed three homologs of AtAARE (Pp3c2_30720V3.1, Pp3c12_21080V3.1, Pp3c7_25140V3.1). Additionally, this search revealed another hit, Pp3c1_2590V3.1. This gene is composed of a single exon and encodes a 131 aa protein which harbors the AARE N-terminal domain (PF19283). However, it lacks a catalytic peptidase domain and is hardly expressed across different culture conditions and tissues[56]. We also did not find any proteomics evidence across several Physcomitrella analyses[36,57–59] for this protein. Therefore, we excluded this gene from further analysis.

We then used phylogenetic reconstruction to investigate the origin of gene duplications within the gene family. As an outgroup, we included the well-characterized rat[16] and human[26] AARE and two isoforms of the Antarctic icefish *Chionodraco hamatus*[27]. Physcomitrella and *F. hygrometrica* share three distinct pairs of orthologs hinting at an expansion in the common ancestor of the two species. Our phylogenetic analysis did not resolve AARE subfamilies across kingdoms (Fig. 1a) and we conclude that the gene family expansions observed in rice and in the Funariaceae are lineage-specific events.

In addition, this analysis reveals a closer relationship between *PpAARE1* and *PpAARE3*, which presumably originate from a more recent gene duplication event, compared to *PpAARE2*. This is supported by the fact that the open reading frames (ORFs) of *PpAARE1* and *PpAARE3* are represented by a single exon whereas the ORF of *PpAARE2* is split across 17 exons, similar to *AtAARE* (Fig. 1b). This is in congruence with a more recent emergence of intron-poor genes in intron-rich families linked to stress response and developmental processes[60] and in line with intron-less orphan Physcomitrella genes as earliest responders to abiotic stress[61].

For the three *PpAARE* genes, several splice variants are annotated, but only two splice variants of *PpAARE1* give rise to distinct protein isoforms (Fig. 1b; PpAARE1_1, PpAARE1_2). Both splice variants are present in Physcomitrella protonema (Fig. 1b, c), according to RT-PCR with splice variant-specific primers (Supplementary Data S1). Likewise, for *AtAARE* two different ORF definitions exist. With Araport11[62], a new version

of the gene model was introduced exhibiting a longer ORF at the 5′ end (Fig. 1b). We detected the full-length transcript via reverse transcription polymerase chain reaction (RT-PCR, Fig. 1c).

For *PpAARE1*, alternative splicing in the 5′ end results in an N-terminal truncated variant whereas the longer non-spliced variant encodes an N-terminal plastid transit peptide (cTP) according to *TargetP2.0*[63]. A cleavage of the transit peptide at the predicted cleavage site (Ala$^{72}$-M$^{73}$, Supplementary Data S2) of PpAARE1 would release exactly the protein encoded by the short splice variant. In contrast, PpAARE2 and PpAARE3 do not harbor any predicted N-terminal targeting signals. Moreover, PpAARE3 is also lacking the WD40 domain that is present in PpAARE1 and PpAARE2 (Fig. 1d).

The extension of the originally annotated ORF of *AtAARE* also encodes a plastid transit peptide (Fig. 1d). To our knowledge, the longer variant of AtAARE has not yet been investigated, whereas the short variant of AtAARE localizes to the nucleus and the cytosol[21]. In agreement with the latter findings, we could predict a nuclear localization sequence (NLS, KKKK) with *LOCALIZER*[64]. Thus, targeting of AtAARE to the cytosol and the nucleus, but also to plastids could be enabled by alternative translation initiation. Likewise, PtAARE harbors a plastid transit peptide and a potential alternative translation initiation site downstream of the predicted cTP cleavage site.

Accordingly, we checked for NLS in PpAARE isoforms and found one (KRRP, Supplementary Data S2) in PpAARE1 and PpAARE3, whereas PpAARE2 has none, further supporting our hypothesis that PpAARE1 and PpAARE3 originate from a relatively recent gene duplication event. Accordingly, alternative splicing also generates two distinct transcripts for AARE1 in rice (OsAARE1, Supplementary Data S2), where one variant encodes a potential plastid transit peptide.

For all other plant species, no plastid targeting sequence was predicted, while the *C. reinhardtii* AARE harbors a mitochondrial targeting sequence (Supplementary Data S2).

## PpAARE1 and AtAARE in mitochondria, chloroplasts, and cytoplasm.

Organellar targeting of AARE has not yet been reported, although AARE activity was observed in plastid-enriched fractions of cucumber[17]. However, chloroplasts, per-oxisomes and mitochondria are major hubs of ROS generation[65], and thus are likely organelles with elevated levels of oxidized proteins. Thus, we investigated whether PpAARE1 and AtAARE would localize to chloroplasts in vivo.

We generated fusion constructs of the PpAARE isoforms and of AtAARE with eGFP for transient expression in Physcomitrella protoplasts. Due to the presence of a predicted plastid targeting peptide for PpAARE1, eGFP was fused in frame to the 3' end of all coding sequences (CDS). Since also peroxisomes are ROS-producing organelles, we used *PlantPredPTS1*[66,67] to check for the presence of C-terminal positioned peroxisomal targeting signals. None of the selected AARE isoforms were predicted to localize to peroxisomes (Supplementary Data S2). Although AtAARE has a C-terminal CKL tripeptide, which is experimen-tally verified to mediate peroxisomal targeting[66,68], the properties of its other C-terminal amino acids most likely prevent peroxisomal targeting. A more recent prediction approach for PTS1-mediated targeting for Arabidopsis proteins[69] further supports this conclusion. Peroxisomal targeting can also be mediated via N-terminal nona-peptides[70,71], but these motifs are also not present within the first 100 aa in any of the selected AARE sequences. In agreement with these predictions eGFP was fused to the 3′ end of the CDSs.

For PpAARE1 three different fusion constructs were assembled. Among these, a fusion of the CDS of the short splice

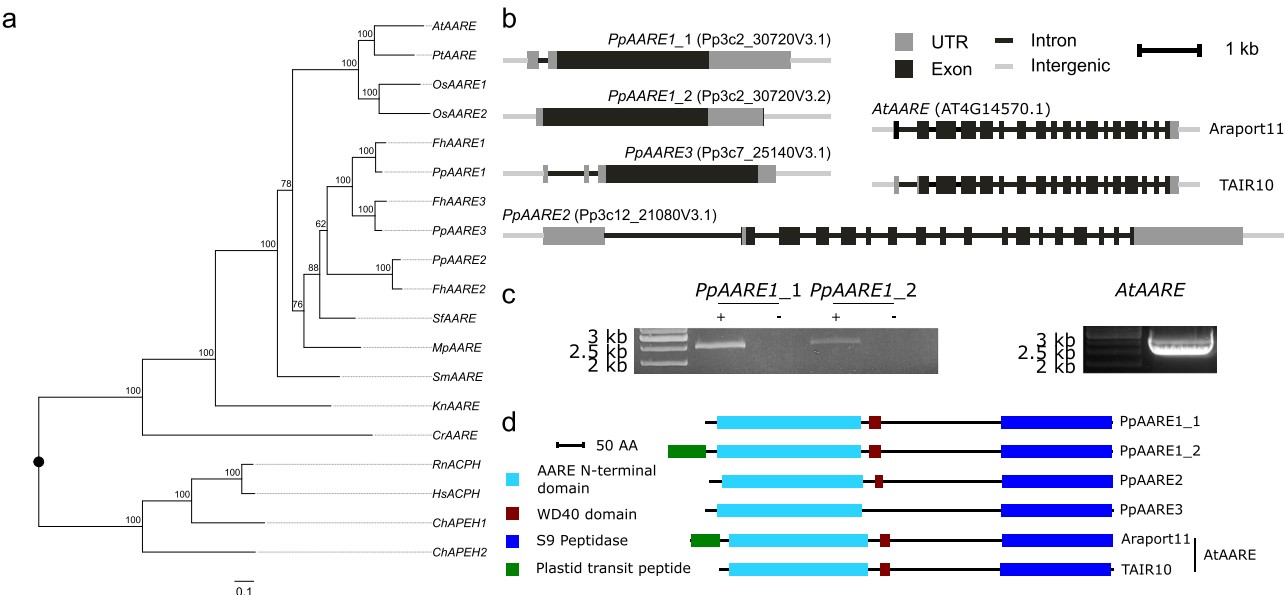

**Fig. 1 Phylogenetic tree for AARE isoforms from selected organisms, gene structures, and proteins domains of *P. patens* and *A. thaliana* AARE isoforms. a** Maximum likelihood tree based on an alignment of multiple *AARE* coding sequences. Duplication events in several species appear to be lineage-specific. Node values reflect percentage confidence values based on 1000 bootstrap replicates. Species abbreviations: At: *Arabidopsis thaliana*; Pt: *Populus trichocarpa*; Os: *Oryza sativa*; Pp: *Physcomitrium patens*; Fh: *Funaria hygrometrica*; Sf: *Sphagnum fallax*; Mp: *Marchantia polymorpha*; Sm: *Selaginella moellendorffii*; Kn: *Klebsormidium nitens*; Cr: *Chlamydomonas reinhardtii*; Rn: *Rattus norvegicus*; Hs: *Homo sapiens*; Ch: *Chionodraco hamatus*. **b** Gene structure of *PpAARE1-3* and *AtAARE*. For *PpAARE1* two splice variants exist. For *AtAARE* two different 5' UTR definitions are present (upper: Araport11[62]; lower: TAIR10[53]). **c** Left: Both splice variants of *PpAARE1* were amplified from complementary DNA (cDNA;+: with cDNA template; -: without reverse transcriptase). Expected amplicon size: *PpAARE1_1*: 2512 bp; *PpAARE1_2*: 2596 bp. Primers were designed to be specific for the splice variants (Supplementary Data S1). Right: the longer open reading frame of *AtAARE* was amplified from cDNA. Expected amplicon size: 2457 bp (Supplementary Data S3). **d** Protein structures showing PFAM-domains for PpAARE1-3 and AtAARE. All isoforms contain an AARE N-terminal domain (PF19283) and a catalytic Peptidase S9 domain (PF00326). PpAARE1, PpAARE2, and AtAARE additionally contain a WD40 domain (PF07676). The long splice variant of *PpAARE1* and the longer open reading frame of *AtAARE* encode a predicted N-terminal plastid transit peptide (cTP). AA amino acid. Cleavable N-terminal sequences were predicted by TargetP2.0[63].

variant (Pp3c2_30720V3.1) and eGFP was cloned, as well as a fusion of the CDS of the longer splice variant (Pp3c2_30720V3.2) and eGFP. Additionally, we cloned a fusion of eGFP and the sequence part in which both variants differ (M$^1$-A$^{72}$, Pp3c2_30720V3.2). This part harbors the plastid transit peptide predicted by *TargetP2.0*. All fusion constructs were expressed under the control of the Physcomitrella Actin5 promoter[57,72] in a pMAV4 plasmid backbone[73].

The PpAARE1 isoform derived from the short splicing variant (PpAARE1_1, Fig. 1b) clearly localized to the cytoplasm (Fig. 2). The same localization was observed for PpAARE2 and PpAARE3 (Fig. 2). Despite a predicted NLS, we did not observe a nuclear localization, either for PpAARE1 or for PpAARE3.

The isoform encoded by the longer splice variant of *PpAARE1* (PpAARE1_2, Fig. 1b) localized to chloroplasts and surprisingly also to mitochondria (Fig. 2). In contrast to the diffuse cytosolic distribution of PpAARE1, specific foci were observed in chloroplasts. To investigate whether the N-terminal sequence differing between the two PpAARE1 variants (M$^1$-A$^{72}$, Pp3c2_30720V3.2) is sufficient to confer dual targeting, we fused this N-terminal sequence 5' to eGFP and observed again dual localization (Fig. 2, PpAARE1_Nt). Full-length PpAARE1 was necessary to localize eGFP to foci within chloroplasts, as the PpAARE1_Nt:eGFP fusion led to a uniform distribution. However, full-length PpAARE1 was also homogeneously distributed throughout the cytoplasm. This indicates the presence of interactors that recruit PpAARE1 to specific sites or complexes within the chloroplasts. Further, we conclude that the N-terminal extension of PpAARE1_2 encodes an ambiguous targeting signal for import into chloroplasts and mitochondria as it is capable of

directing the fusion protein simultaneously to both organelles. Based on this data, PpAARE1 is targeted to three organelles: chloroplasts, mitochondria and the cytosol. To independently scrutinize these findings, we measured AARE enzyme activity in the organelles. Using organelle fractionation as previously described[74], we detected AARE activity in chloroplasts, the cytosol, and mitochondria, although to a lesser extent in the latter (Fig. S3). Thus, in vivo localization of fusion proteins and enzyme measurements after cell fractionation independently confirm the predicted triple localization of AARE in Physcomitrella.

Simultaneous localization of proteins to chloroplasts and mitochondria can be mediated *via* ambiguous targeting signals which are recognized by both translocation machineries. We evaluated whether *ATP2*[75], a tool for the prediction of ambiguous targeting, would recognize PpAARE1 but this was not predicted to be dually targeted. In contrast, AtAARE was predicted to be dually targeted via an ambiguous signal. Thus, we cloned the analogous three fusion constructs for AtAARE and investigated the subcellular localization of their encoded proteins accordingly.

The AtAARE isoform translated from the shorter ORF (AtAARE, Fig. 1b) localized to the cytoplasm (AtAARE_SV, Fig. 2), as observed for PpAARE1_1. This result is partially in agreement with Nakai et al.[21] since we could not observe nuclear localization. Using the fusion construct of the longer AtAARE variant, we observed clear dual targeting of the protein to chloroplasts and mitochondria (AtAARE_LV, Fig. 2), as observed for PpAARE1_2. Here, the eGFP signal was distributed homogeneously in the chloroplasts, in contrast to the foci of PpAARE1:eGFP.

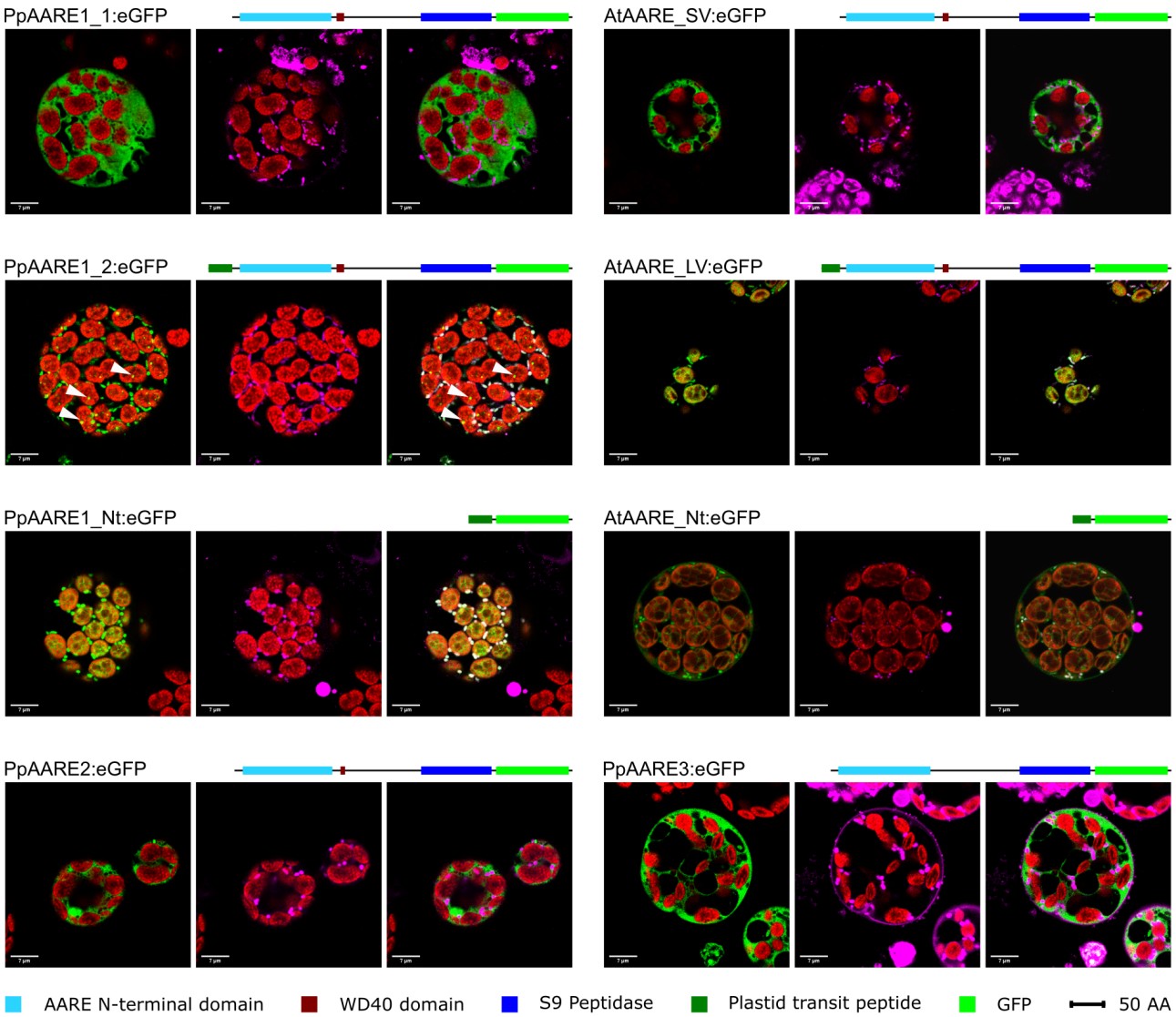

**Fig. 2 Confocal microscopy images showing the localization of PpAARE isoforms and AtAARE with C-terminal fused eGFP in Physcomitrella protoplasts.** Fusion proteins with domain structures are depicted. PpAARE1_1, PpAARE2, and PpAARE3 localize to the cytoplasm. PpAARE1_2 localizes to specific foci in plastids (white arrows) and to mitochondria. The N-terminal extension of PpAARE1_2 encoding a predicted plastid transit peptide (PpAARE1_Nt) directs eGFP to plastids and mitochondria. The short variant of AtAARE (SV) localizes to the cytoplasm. The long variant (LV) localizes to plastids and mitochondria. The N-terminal extension of the long variant of AtAARE localizes to plastids and mitochondria. Left image: chlorophyll autofluorescence (red) and eGFP (green). Middle image: chlorophyll autofluorescence (red) and MitoTracker™ (magenta). Right image: chlorophyll autofluorescence (red), eGFP (green), Mitotracker™ (magenta) and co-localization of eGFP and MitoTracker™ (white). Bars = 7 μm.

Next, we cloned only the N-terminal sequence differing between both variants ($M^1$-$A^{55}$, longer ORF definition, Fig. 1b) and fused it to eGFP. In order to investigate whether the exact N-terminal difference between the two AtAARE variants would be sufficient for targeting, the $M^{56}$ (same as $M^1$ in the shorter variant), which is the P1 aa at the predicted cleavage site, was deleted. Using this construct, the eGFP signal localized to chloroplasts and mitochondria (AtAARE_Nt, Fig. 2). The signal within chloroplasts was homogeneously distributed, similar to the longer AtAARE variant. Thus, we conclude that the N-terminal extension of both long PpAARE and AtAARE variants is sufficient for dual targeting of proteins in vivo. Intriguingly, the longer variant of AtAARE localized exclusively to chloroplasts and mitochondria although alternative translation initiation should be possible. This is interesting as alternative translation initiation is also possible in the longer splice variant of *PpAARE1* (*PpAARE1_2*). In the latter also, the fusion protein localizes

exclusively to chloroplasts and mitochondria, which excludes the possibility of an alternative translation initiation, at least in protoplasts. There are numerous transcripts in mammals where translation of an upstream positioned ORF suppresses the translation of the downstream main ORF[76]. A similar scenario is conceivable in Physcomitrella. However, it remains unclear how and if translation from the internal start codons is controlled. It is also possible that factors controlling alternative translation initiation of AtAARE are absent in Physcomitrella, at least in a spatiotemporal manner, or they might only be triggered in specific physiological situations. According to our data, the translation of the two variants of PpAARE1 is mainly controlled by alternative splicing and not by alternative translation initiation.

In summary, PpAARE1 and AtAARE localize to three subcellular compartments via an ambiguous targeting signal. In contrast, PpAARE2 and PpAARE3 localize solely to the cytoplasm.

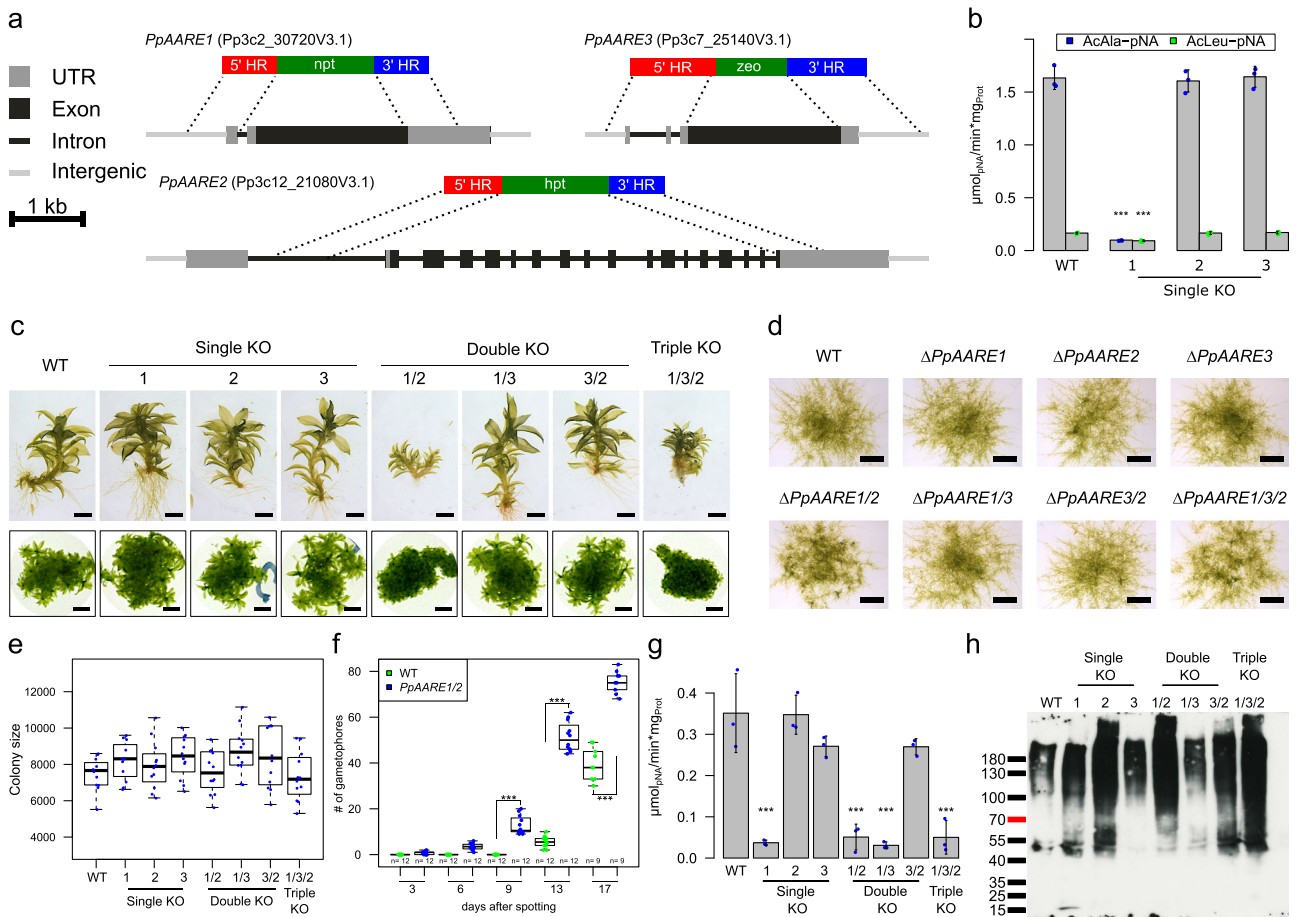

**Fig. 3 Phenotype analyses, AARE activity and level of oxidized proteins in Physcomitrella WT and KO mutants. a** Each isoform was knocked out using a different selection marker. The selection markers confer resistance against G418 (npt), hygromycin (hpt) and zeocin (zeo), respectively. Homologous regions (HR) were chosen in order to remove the full CDS of the corresponding gene upon integration of the selection marker. **b** Mean AARE activity on AcAla-pNA in Physcomitrella protonema with standard deviation ($n =$ three biological replicates). Significance levels are based on a one-way Anova and subsequent post-hoc test (***$p < 0.001$) and indicate differences compared to WT. **c** Phenotypes of gametophores of WT and the different KOs cultivated on KnopME for 4 months. All colonies were grown on the same plate. Upper panel: Bar = 0.5 mm; lower panel: Bar = 2 mm. **d** Protonema colonies grown on KnopME with 0.02% glucose taken from suspension culture. Bar = 1 mm. Images were taken 9 days after spotting. **e** Box plot showing the distribution of colony areas ($n = 12$ colonies per box plot; colony images taken 7 days after spotting). No significant difference of colony areas between WT and the KOs was observed (tested via one-way Anova). **f** Box plot showing the number of gametophores per colony and day of WT and $\Delta PpAARE1/2$. Counting was performed at indicated days after spotting protonema suspension culture on solid medium (KnopME with 0.02% glucose). Indicated significance levels are based on a two-way Anova with subsequent post-hoc test (***$p < 0.001$). The box plots (**e, f**) depict the mean (horizontal bold line) of the data, the interquartile range (box) and the 1.5x interquartile range (whiskers). **g** Mean AARE exopeptidase activity on AcAla-pNA in gametophores. Depicted is the mean of three independent colonies per line with standard deviations. All colonies were cultivated on the same plate (Fig. S7a). Significance levels are based on one-way Anova and indicate differences compared to WT (***$p < 0.001$). **h** Levels of oxidized proteins in gametophores of WT and the different KOs (Coomassie-stained loading control is shown in Fig. S7b). The analysis was repeated three times (additional blots available in Fig. S7c, d). Detection of oxidized proteins was performed with the OxyBlot™ *Protein Oxidation Detection Kit* (Merck).

**Double knockout of *PpAARE1/2* reduces lifespan.** Null mutants of *AARE* have not been described in any organism, and the biological role of this protease apart from its catalytic function remained unclear. Hence, we created *AARE* knockouts in Physcomitrella by deleting the full CDS of each gene *via* gene targeting (Fig. 3a) according to[77]. To enable subsequent knockout (KO) of other AARE genes, different selection markers were used. Since three distinct AARE genes exist in Physcomitrella which result in cytosolic proteases, we generated all possible combinations of double KOs and triple KOs to avoid potential compensation of the loss of function.

Plants surviving the selection procedure were screened via PCR for the absence of the respective CDS and correct integration of the KO construct in the target locus (Fig. S4a). Additionally, the number of genomic integrations of the KO construct was

measured *via* quantitative PCR (qPCR) as described[78] (Fig. S4b). At least three independent lines were identified for all single, double and triple KOs (Fig. S4c–i) and a line with only a single integration in the genome was detected for each KO (j-p, line numbers with stars). Further, haploidy of all lines was confirmed *via* flow cytometry (Fig. S5a–c) as described[79]. These precautions were made as the transformation procedure may generate plants with multiple integrations[80], possibly leading to off-target effects. Further, the transformation procedure may lead to diploid plants with altered gene expression[81]. We used haploid lines with a single integration of the KO construct (Fig. S4) for subsequent experiments.

Typically, AARE exopeptidase activity is assayed *via* N$^\alpha$-acetylated amino acids like AcAla or AcMet coupled to a reporter such as para-nitro-anilide (pNA) or 7-amido-4-methylcoumarin

(AMC). From these, AcAla-pNA was tested for several eukaryotic AAREs[22,26], including AtAARE[17]. Here, we analyzed the impact of AARE loss of function on the activity towards AcAla-pNA and AcLeu-pNA. The latter is a substrate of bacterial and archaeal AARE isoforms but also eukaryotic isoforms exhibit cleavage activity on this substrate[16,22–25]. On the single KO level, the exopeptidase activity on both substrates was significantly reduced in the ΔPpAARE1 mutant whereas the single KO of the other isoforms did not affect the activity (Fig. 3b). This strong impact of PpAARE1 on the exopeptidase activity was consistent across all transgenic mutant lines (Fig. S6).

One important step in Physcomitrella development is the transition from protonema to gametophores, a developmental progression regulated among others by plant age and the nutrient status[82]. The single KOs were phenotypically inconspicuous on the gametophore level (Fig. 3c). In contrast, gametophores of ΔPpAARE1/2 and ΔPpAARE1/3/2 were severely stunted and colonies were denser compared to wild type (WT) or the other mutants (Fig. 3c). This growth effect is restricted to gametophores since protonema growth on solid medium did not differ between WT and the KOs (Fig. 3d, e). Intriguingly, ΔPpAARE1/2 and ΔPpAARE1/3/2 mutants showed accelerated developmental transition, as they developed gametophores from protonema earlier (Fig. 3d). Since other KO lines did not show this effect, we attributed this to the double KO of PpAARE1/2 and performed a quantitative comparison with WT. Here, gametophores were already observed after 6 days in the double KO of PpAARE1/2, while in WT a similar number of gametophores per colony was observed only after 13 days (Fig. 3f). Consequently, the double KO of PpAARE1/2 causes accelerated developmental progression but gametophores remained ultimately smaller than in WT (Fig. 3c). These effects are not linked to AARE exopeptidase activity since the exopeptidase activity in gametophores was significantly reduced in all lines with a KO of PpAARE1 (Fig. 3g), which mimics the activity profile in protonema (Fig. S6).

To analyze AARE endopeptidase activity, we assessed the total levels of oxidized proteins in gametophores. In this assay, protein carbonyl groups derived from oxidation are derivatized with 2,4-dinitrophenylhydrazine (DNPH) to 2,4-dinitrophenylhydrazone (DNP). This irreversible modification is then recognized on Western blots by a primary anti-DNP antibody. Since DNPH can also react with oxidation states of cysteine side chains[83], this assay detects not only protein carbonylation but general protein oxidation.

With this assay we found that PpAARE2 had the strongest impact on the level of oxidized proteins in gametophores (Fig. 3h) and thus is not linked to exopeptidase activity. This was consistently observed in three independent analyses (Fig. 3h, Fig. S7). Apparently, PpAARE3 does not have any impact on exopeptidase and endopeptidase activity in gametophores under standard conditions (Fig. 3g, h).

Taken together, PpAARE1 predominantly acts as exopeptidase, while PpAARE2 predominantly acts as endopeptidase, and only the simultaneous loss of both activities in the double knockout mutants has the severest phenotypical consequences.

We found another remarkable difference between WT and mutants with a double KO of PpAARE1/2 in older plants. After 5 months of cultivation, ΔPpAARE1/2 and ΔPpAARE1/3/2 were only viable at the tip of the gametophores (Fig. 4a), whilst most of the colony was already dead. In contrast, gametophores of WT and the other KOs were fully viable. After 8 months, ΔPpAARE1/2 and ΔPpAARE1/3/2 were already dead, in contrast to WT and the respective parental lines, which only showed some dead gametophores (Fig. 4b, c).

In summary, mutants with a double KO of PpAARE1/2 exhibit accelerated developmental transition from protonema to gametophore (Fig. 3f), while size and life span of gametophores is strikingly reduced (Figs. 3c and 4a–c). In contrast, these effects are not visible in ΔPpAARE1/3. Therefore, these ageing phenotypes are linked to the concurrent loss of major AARE endopeptidase and exopeptidase activity.

**Distinct in vivo interactions of PpAARE isoforms**. In different organisms, AARE forms different homomeric complexes such as dimers[14], tetramers[17], or hexamers[24]. Thus, we analyzed whether the PpAARE isoforms can interact with each other. Previously, all three isoforms were identified, although the protein modification used for pulldown (N-terminal arginylation) was only identified on PpAARE1[36]. This gave rise to two hypotheses: First, PpAARE2 and PpAARE3 are also targets for N-terminal arginylation, but modified peptides were not identified for these isoforms. Second, the isoforms interact in complexes which were pulled down due to the N-terminal arginylation of PpAARE1. We generated Citrine fusion lines for each isoform via in-frame tagging at the native locus (knock-in, Fig. 5a). The original stop codons of the respective PpAARE CDS were deleted. The Citrine-tag was intended for two different analyses: First, it should enable in vivo monitoring of PpAARE isoforms expressed from the native promoter, and second, it is a valid tag for co-immunoprecipitation (Co-IP) via commercially available trap-beads.

In plants with a detectable fusion transcript (Fig. S8a–c) the presence of the target protein was checked via IP and subsequent MS analysis. For PpAARE3:Citrine lines, we detected transcripts in one line and obtained only insufficient coverage and intensity at the MS across several Co-IPs. Thus, these Physcomitrella lines were excluded from further analysis. The co-precipitation of other PpAARE isoforms with the respective bait isoforms was observed in test IPs (Fig. S8d, e) confirming previous MS-Data[36]. All plants harboring the Citrine fusion were phenotypically inconspicuous (Fig. S8f) and haploid (Fig. S8g). Although the fusion proteins were detected in two independent lines for each of the two isoforms (PpAARE1, PpAARE2), we could not observe any Citrine signal within Physcomitrella protonemata or gametophores, probably due to the low abundance of the PpAARE isoforms. Nevertheless, MS intensities and sequence coverage enabled quantitative Co-IPs. The MS data have been deposited in PRIDE[84,85] with the accession codes PXD033854 and PXD038742.

When targeting PpAARE1:Citrine, both other isoforms appeared as significant interacting partners (Fig. 5b, $p < 0.01$, FDR = 0.01). In a reciprocal Co-IP targeting PpAARE2:Citrine only, PpAARE1 appeared as significant interacting partner. PpAARE3 was not detected in this pulldown. Although lacking a reciprocal IP targeting PpAARE3:Citrine, the data show that PpAARE1 interacts with PpAARE2 and PpAARE3, whereas PpAARE2 only interacts with PpAARE1. Consequently, there are distinct interactions of PpAARE1 with PpAARE2 and PpAARE3 in vivo, possibly resulting in cytosolic heteromeric AARE complexes in Physcomitrella.

**AARE affects bolting time in Arabidopsis**. In Physcomitrella three AARE genes exist, and the concerted action and interaction of the enzymes affect plant ageing. To evaluate if this is an evolutionary conserved function, we analyzed the situation in Arabidopsis. Here, it was known that silencing of the single AARE gene leads to an accumulation of oxidized proteins, whereas overexpression did not affect their levels[21]. To gain deeper insights, we screened for available Arabidopsis T-DNA mutants.

We identified two T-DNA insertion lines (SALK_071125 and GK-456A10) in the T-DNA Express database at the SIGnAL website (http://signal.salk.edu). SALK_071125 (s68) has a T-DNA

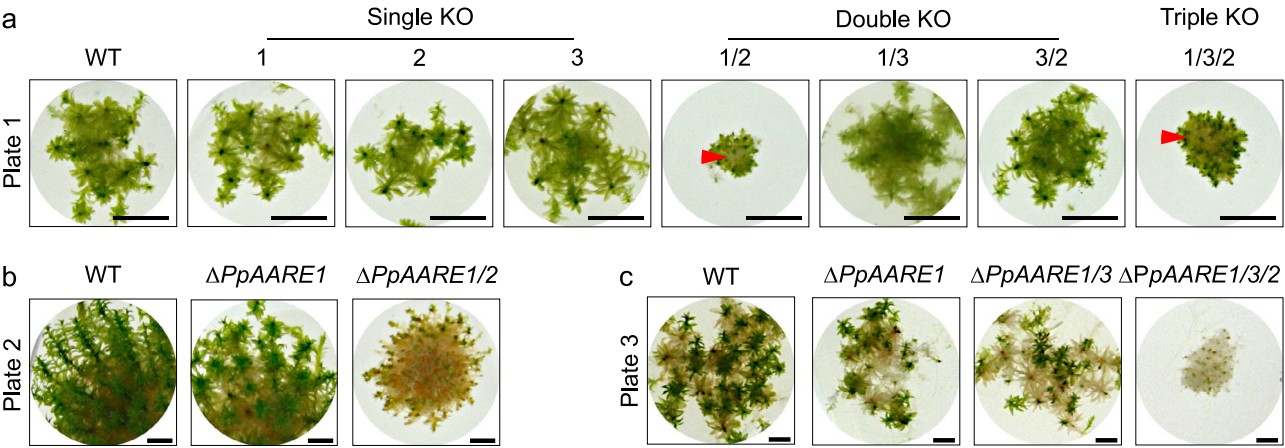

**Fig. 4 Physcomitrella gametophore colonies of varying age. a** Colonies of all representative knockout mutants after 5 months on solid medium (KnopME). Bar = 5 mm. Gametophores of ΔPpAARE1/2 and ΔPpAARE1/3/2 are only viable at the tip, whereas plant material at the base (red arrow) is dead. **b** Colonies after 8 months on solid medium. ΔPpAARE1 is the parental line for ΔPpAARE1/2. Gametophores of WT and ΔPpAARE1 are still viable whereas ΔPpAARE1/2 gametophores are mostly dead. Bar = 2 mm. **c** Colonies after 8 months on solid medium. ΔPpAARE1 is the parental line for ΔPpAARE1/3 and ΔPpAARE1/3 is the parental line for ΔPpAARE1/3/2. Colonies of WT, ΔPpAARE1 and ΔPpAARE1/3 still have viable gametophores, whereas ΔPpAARE1/3/2 gametophores are dead. Bar = 2 mm.

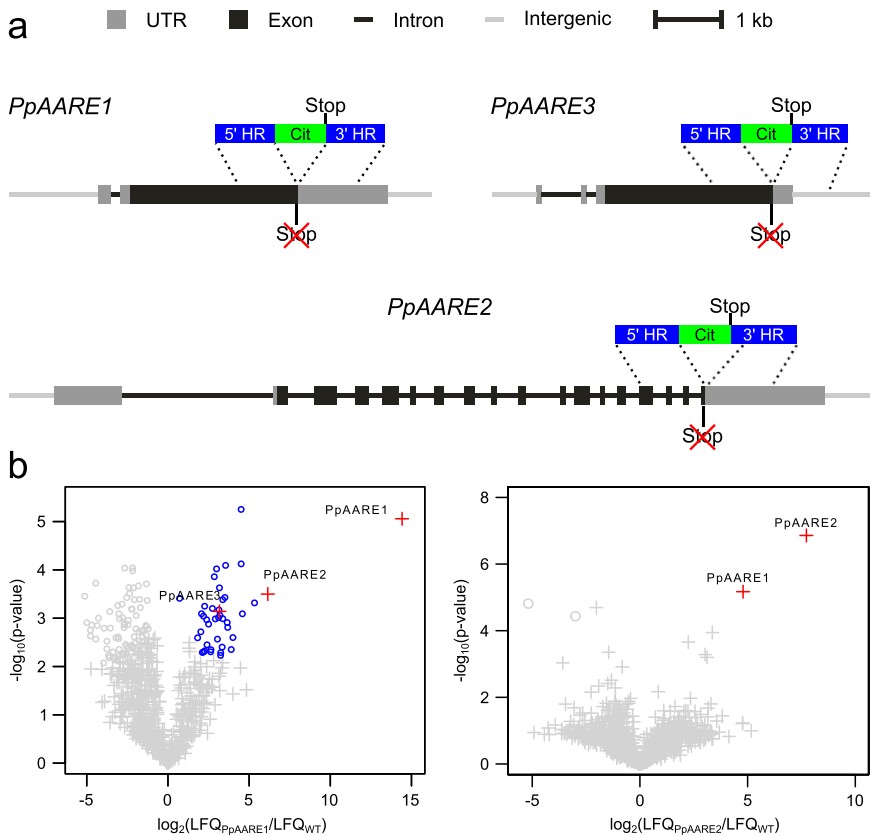

**Fig. 5 Reciprocal co-immunoprecipitation (Co-IP) with Citrine-tagged PpAARE isoforms. a** Tagging of PpAARE isoforms was realized by in-frame fusion with a linker-*Citrine* CDS at the respective native locus *via* homologous recombination. Original stop codons were deleted. **b** Volcano plots showing the result of the Co-IPs against each of the PpAARE:Citrine fusion proteins. Left panel: Pulldown of PpAARE1:Citrine. Right panel: pulldown of PpAARE2:Citrine. Co-IP was performed with GFP-Trap Magnetic Particles M-270 (Chromotek) and protonema from suspension culture. Log$_2$ ratios of normalized label-free quantitation values (LFQ) are plotted against –log$_{10}$ of adjusted *p*-values. Proteins significantly enriched in the Citrine-tagged pulldown are shown in blue circles (*p* < 0.01, FDR = 0.01). Significantly enriched PpAARE isoforms are depicted as red crosses.

insertion in an intron at the 5′ UTR, whereas the T-DNA insertion in GK-456A10 (GK) maps to an intron in the region encoding the catalytic domain (Fig. 6a). We obtained these mutants, identified homozygous populations of GK-456A410

containing the T-DNA insertion by their resistance to sulfadia-zine, and confirmed their genotype by PCR. In the case of the Salk line (s68), homozygous plants had lost their resistance to kanamycin, but we confirmed their genotype by PCR.

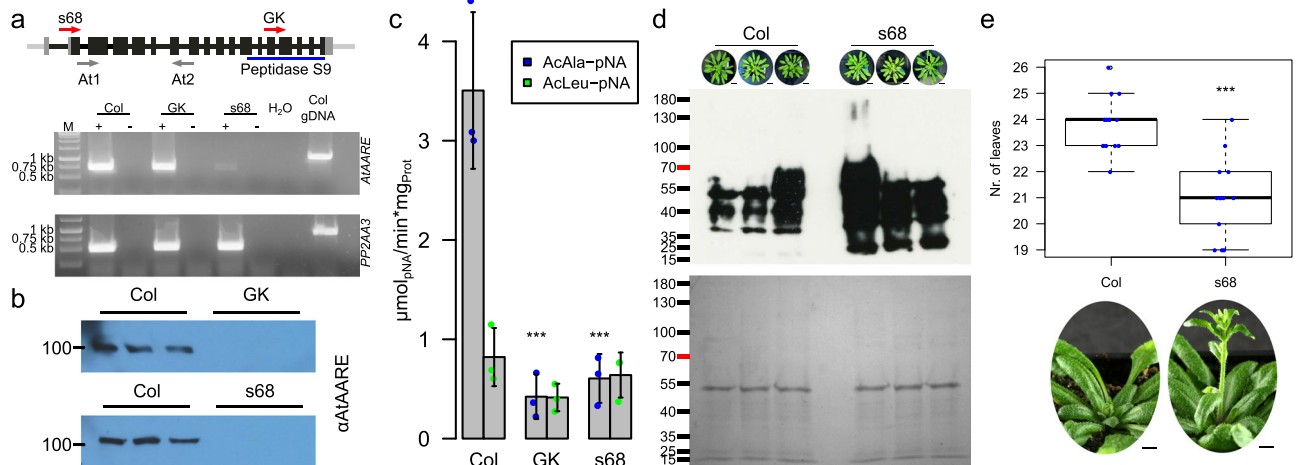

**Fig. 6 Molecular and phenotypical characterization of _A. thaliana_ T-DNA mutant lines. a** Structure of the AtAARE gene and positions of T-DNA insertions (red arrows indicate positions of T-DNA insertion, s68 (SALK_071125) and GK (GK-456A10)) and primers (At1, At2) used for RT-PCR (below). Transcription analysis of _AtAARE_ was performed by RT-PCR in WT (Col), s68, and GK plants. Negative controls without reverse transcriptase (-), a water control (H₂O) and genomic DNA control (gDNA) are indicated. Expected amplicon sizes: _AtAARE_: cDNA 739 bp, gDNA: 1204 bp; _PP2AA3_ (reference gene): cDNA: 517 bp, gDNA: 911 bp. Primers are listed in Supplementary Data S3. **b** Western blot analysis on extracts of the two T-DNA mutants and WT probed with a polyclonal anti-AARE antibody. In both T-DNA lines AtAARE is not detectable. Full blot images are available in Fig. S10. **c** Mean exopeptidase activity in _A. thaliana_ extracts on AcAla-pNA and AcLeu-pNA with standard deviation ($n = 3$ biological replicates). Significance levels are based on one-way Anova and subsequent post-hoc test (***$p < 0.001$) and indicate differences compared to WT (Col). **d** Detection of oxidized proteins in WT (Col) and _AtAARE_ mutant (s68). From three independent plants one exterior rosette leaf was taken after cultivation for 8 weeks in short-day conditions (8 h light/ 16 h dark). Bars correspond to 2 cm. Protein loading control by Coomassie staining is depicted below. **e** Box plot of the bolting analysis of WT (Col, $n = 14$) and _AtAARE_ mutant plants (s68, $n = 13$). The box plot (**e, f**) depicts the mean (horizontal bold line) of the data, the interquartile range (box) and the 1.5x interquartile range (whiskers). Outliers are depicted as white dots. Significance level is based on a one-way Anova and subsequent post-hoc test (***$p < 0.001$). Exemplary plants (45-day-old plants grown in long-day conditions, 16 h light/8 h dark) are depicted below the box plot. Bars correspond to 1 cm.

Homozygous plants of both genotypes were back-crossed with WT Col-0, and brought to homozygosis for subsequent experiments. Primers for screening and validation of both T-DNA lines are listed in Supplementary Data S3. Additionally, we analyzed _AARE_ gene expression _via_ RT-PCR. The transcript was still detectable in both lines (Fig. 6a), although very reduced in s68, while the protein was not detectable via Western blot (Fig. 6b). Surprisingly, in WT AARE was detected at around 100 kDa, although the estimated molecular weight is ~84 kDa (90 kDa for the longer ORF variant without cleavage of the plastid targeting peptide) which indicates the presence of posttranslational modifications. Phenotypically, neither seedlings nor adult mutant plants showed obvious deviations from WT (Col, Fig. S9).

Next, we assayed AARE exopeptidase function in Arabidopsis WT and the two mutants. The exopeptidase activity on AcAla-pNA was significantly reduced in both T-DNA mutants. In contrast, the activity on AcLeu-pNA did not change significantly, although a slight reduction was observed in the GK line (Fig. 6c). Thus, in agreement with Western blot and activity data, the remaining transcripts detected _via_ RT-PCR (Fig. 6a) are not translated to functional enzymes, suggesting complete loss of AARE function in the mutants. Based on this characterization, we concentrated on T-DNA mutant s68.

Subsequently, we assayed the AARE endopeptidase activity in Arabidopsis. The most striking feature reported were increased levels of oxidized proteins in AARE-silenced Arabidopsis plants[21] which is in line with our findings in Physcomitrella AARE mutants. To corroborate this, we investigated levels of oxidized proteins in the Arabidopsis T-DNA mutant (s68) in comparison to WT (Col) cultivated under short day conditions. In both genotypes, the distribution of oxidized proteins was different to Physcomitrella gametophores (Fig. 6d). Using the same extraction protocol, oxidized proteins in Arabidopsis were mainly of lower

molecular weight, whereas in Physcomitrella they were mainly at higher molecular weight (Fig. 3h). Despite these size differences, the level of oxidized proteins was higher in the Arabidopsis mutant than in WT (Fig. 6d), mimicking the situation in Physcomitrella. Together, we found AtAARE exopeptidase and endopeptidase activity to be reduced in our experiments.

Oxidized proteins accumulate during vegetative growth of Arabidopsis and are depleted at the transition to bolting[86]. It was not clear from this study whether the level of oxidized proteins is a signal for bolting, or if the reset of protein oxidation at the bolting transition is a side-effect of enhanced protein turnover during this process. To address this, we assessed the bolting behavior of Arabidopsis WT and mutant plants and found that bolting in mutant plants differed significantly from WT. In WT, bolting started at $24 \pm 1$ rosette leaves whereas it started at $21 \pm 2$ rosette leaves in the mutants (Fig. 6e). Consequently, this accelerated developmental transition in the mutants correlates with enhanced levels of oxidized proteins.

## Discussion
A universal definition of ageing is difficult due to strongly differing characteristics between and within the domains of life. In humans, the term ageing is inherently linked to degenerative diseases whereas plant ageing is considered as the progression of developmental transitions from seeds to senescence. Nevertheless, a common feature of ageing is the execution of a genetic program that controls growth, development and maturation. In turn, the progression of this genetic program depends on the metabolic state and environmental factors. Among the molecules responsible for ageing are ROS. Their involvement in ageing and diseases was first postulated by Harman[87] and was extended by considering mitochondria as central sources[88]. In humans,

mitochondrial malfunction in line with increased ROS levels is a central determinant of ageing and associated pathologies[89]. In plants, mitochondrial ROS increase during ageing of seeds[90] and are major determinants of germination capacity[91]. In photosynthetic tissues chloroplasts are the major source of ROS and their levels increase during ageing[92–94]. Plants cannot escape from stress situations that increase ROS production to detrimental levels, and despite several layers of ROS defense, oxidized proteins constitute the major share of modified molecules under stress. Consequently, protein oxidation and subsequent aggregate deposition are hallmarks of ageing[95,96]. The degradation of artificially oxidized proteins by AARE has been demonstrated repeatedly, and silencing of *AtAARE* in turn increased the levels of oxidized proteins[19–22].

However, the contribution of AARE to the progress of ageing remained elusive, although several studies associate AARE function to age-related diseases[97–99]. Here, we identified three *AARE* genes in Physcomitrella. This gene family expansion is an outstanding feature of the moss family Funariaceae, as most organisms we analyzed contain only a single *AARE* gene, and only a few have two. We analyzed these three isoforms in Physcomitrella and compared selected features to the single isoform in Arabidopsis. Our data reveal specific functions in age-related developmental transitions and life span determination.

Surprisingly, we found triple localization of AtAARE and PpAARE1 to chloroplasts, mitochondria and the cytosol in vivo, suggesting a functional role for AARE in these cellular compartments. In Physcomitrella, the triple localization is mediated via alternative splice variants and in Arabidopsis likely via alternative translation initiation. Although there have been indications of AtAARE being associated to chloroplasts[17,21], there was no clear evidence for organellar targeting of this protease until now. It is remarkable that the triple localization of AARE is evolutionary conserved between Arabidopsis and Physcomitrella, although likely executed via different molecular mechanisms. This suggests an essential and evolutionary conserved function of AARE activity in the cytoplasm, chloroplasts and mitochondria. As mosses and seed plants diverged more than 500 million years ago[100], this is a deep evolutionary conservation.

Previously, AARE exopeptidase activity was observed in cucumber chloroplasts[17], and AARE peptides were found in proteomes of Arabidopsis chloroplasts[101] and mitochondria[102]. In contrast, AARE was not identified in plastid or mitochondrial proteomes of Physcomitrella[57,59]. We found AARE exopeptidase activity in chloroplasts and mitochondria of Physcomitrella. Chloroplasts are a major source of ROS in photosynthetic tissues exposed to light, whereas mitochondria are the major source of ROS in non-photosynthetic tissues or in the dark[93,94,103,104]. Until now, it remained unresolved how plants deplete oxidized proteins from these organelles.

In yeast and mammals, the ATP-dependent LON and AAA proteases are involved in clearance of misfolded and oxidized mitochondrial proteins. Intriguingly, mutants of plant homologs of LON proteases did not show clear effects on the levels of oxidized proteins, but AAA-type FTSH proteases may play a role[105]. Nevertheless, stressors such as heat, drought or intense light compromise photosynthesis[106,107] and mitochondrial respiration[108], leading to a depletion of ATP and to mitochondrial dysfunction[109]. In turn, energy supply for ATP-dependent defense systems such as heat-shock proteins and AAA-type proteases is severely compromised, leaving the question unanswered how oxidized proteins in chloroplasts and mitochondria can be cleared. Because AARE is an ATP-independent protease, our data suggest that organellar-targeted AARE may act as an ATP-independent defense to prevent or attenuate protein aggregation in the major ROS-producing organelles.

Based on localization prediction, 70% of our selected plant species possess one AARE isoform that localizes to either plastids or to mitochondria. Whether dual targeting via ambiguous targeting also occurs in these species remains to be experimentally validated. Further, we do not exclude that the remaining species also have organellar AARE isoforms, because our predictions may be compromised by incomplete gene models. Such incomplete AARE gene models without a transit peptide were present in earlier genome annotations of Arabidopsis and Physcomitrella[41,53], whilst complete gene models with transit peptides were only introduced with later versions[39,62].

It is not yet clear how translation of both variants in Arabidopsis is regulated, but a recent study highlights the importance of alternative translation initiation in shaping different subcellular localizations or functions of proteoforms[110]. This mechanism is also present in Physcomitrella, where dual targeting of FtsZ isoforms to the cytoplasm and chloroplasts is enabled via alternative translation initiation[111]. Thus, localization of PpAARE1 to the cytoplasm is also possible from the longer splice variant. Alternative translation initiation of *AtAARE* is further evidenced by proteomics data (www.peptideatlas.org/builds/arabidopsis/)[112]. Here, we found evidence for N-terminal acetylation of M[56] which is the initiator methionine of the short variant (PXD012708[113]).

A reporter fusion of the shorter *AtAARE* ORF was observed in the cytoplasm and in the nucleus[21]. We did not detect a nuclear localization of any AARE isoform. Using *LOCALIZER*[64] we identified an NLS in AtAARE, in PpAARE1 and PpAARE3, but not in PpAARE2. In contrast, the human AARE homolog HsACPH does not have a predictable NLS, but nuclear import is mediated via interaction with XRCC1 under stress, where it acts in DNA-damage repair[114]. Similarly, a nuclear localization of AARE might also occur under stress *in planta*.

In Physcomitrella, AARE1 is the dominant exopeptidase, whereas AARE2 acts predominantly as endopeptidase, and the operation mode of AARE3 remains unresolved. Crystal structures from bacterial and archaeal AAREs revealed two possible entrances for substrates to the catalytic centers[14,115] but the mode of substrate entry is not fully understood. Although the quaternary arrangements of subunits differ between species[115,116], the secondary structure arrangement is conserved across kingdoms and specific subunit interactions (multimerization) are likely a mechanism to mediate substrate specificity and modulate activity. This could be further used as a switch between endopeptidase and exopeptidase activity and additionally enable access of the catalytic center via structural re-arrangements. Accordingly, the interaction between the distinct PpAARE isoforms may modify substrate specificity and activity.

Our data indicate a partial compensation between PpAARE1 and PpAARE2. Both single KO mutants are phenotypically inconspicuous under normal conditions, although a significant reduction in exopeptidase activity was observed in ΔPpAARE1. PpAARE2 had the strongest effect on the accumulation of oxidized proteins in gametophores on the single isoform KO level, whereas the double KO of *PpAARE1* and *PpAARE2* showed the most striking effect in all lines as accelerated transition from protonema to gametophores. This phenotype is similar to mutants with disturbed auxin transport[117]. However, colony growth in these mutants was reduced which is different from our double KO mutant (*PpAARE1/2*). Intriguingly, this double KO results in stunted gametophores and a reduced life span. This phenotype partially resembles a loss-of-function mutant of a central component of autophagy (PpATG3) in Physcomitrella[118]. In ΔPpATG3, gametophores exhibit a reduced life span and colonies are smaller than in WT. In contrast, gametophore size was not affected. Further, photosynthetic capacity in ΔPpATG3 was also reduced, an effect which is apparently not caused by

AARE depletion[21] and *PpAARE* genes were not differentially expressed in Δ*PpATG3*[118]. We conclude that the reduced life span observed in Δ*PpAARE1/2* and Δ*PpAARE1/3/2* is not due to an impaired autophagy system causing nitrogen starvation. This is in line with data which opposes autophagy at the onset of senescence in Arabidopsis[119].

In mammals, elevated levels of oxidized proteins are associated with age-related pathologies, such as Alzheimer's disease, diabetes and different types of carcinomas[120]. If proteolytic clearance fails, further accumulation of oxidized proteins causes protein aggregation, which is a hallmark of ageing in animals[96,121,122]. A connection between protein oxidation and ageing was less well studied in plants. Plastid ROS levels increase during ageing[92], which is in line with strong oxidation of plastid proteins in ageing leaves[86]. Likewise, protein oxidation marks the developmental transition between vegetative growth and flowering in Arabidopsis[86]. Physcomitrella mutants Δ*PpAARE1/2* and Δ*PpAARE1/3/2* showed accelerated developmental transition from protonema to gametophores, reduced life span and increased levels of oxidized proteins as signs of accelerated ageing. This is supported by the fact that gametophore tips, which is younger tissue, are viable longer than the older stems in both mutants. In the Arabidopsis *AARE* T-DNA mutants we found increased levels of oxidized proteins under normal cultivation conditions and an accelerated developmental transition, in this case premature bolting. These findings suggest an evolutionary conserved connection between protein oxidation and ageing.

We provide here a detailed analysis of *AARE* genes in the plant lineage and an in-depth analysis of AARE localization and function in the moss Physcomitrella and the annual angiosperm Arabidopsis. *AARE* loss-of-function mutants have not been described for any organism so far. We generated and analyzed such mutants and describe a connection between AARE function, aggregation of oxidized proteins and plant ageing, including accelerated developmental progression and reduced life span. Our findings complement similar findings in humans and animals where AARE malfunction is associated with protein aggregation and age-related pathologies.

To solidify the role of AARE in ageing in different life forms, particularly in plants with contrasting maximum life spans and in animals of different complexity, loss-of-function mutants should be established and analyzed in selected model species. Likewise, a deeper understanding of AARE function in human diseases is desirable. Together, such analyses may contribute to a unified concept of ageing in different life forms.

## Methods

**Cultivation of Physcomitrella**. Physcomitrella WT (new species name: *Physcomitrium patens* (Hedw.) Mitt[123].); ecotype "Gransden 2004" and AARE KO lines were cultivated in Knop medium[124] supplemented with microelements. Knop medium (pH 5.8) containing 250 mg/L $KH_2PO_4$, 250 mg/L KCl, 250 mg/L $MgSO_4 \times 7\ H_2O$, 1,000 mg/L $Ca(NO_3)_2 \times 4\ H_2O$ and 12.5 mg/L $FeSO_4 \times 7\ H_2O$ supplemented with 10 mL per litre of a microelement (ME) stock solution[125,126] (309 mg/L $H_3BO_3$, 845 mg/L $MnSO_4 \times 1\ H_2O$, 431 mg/L $ZnSO_4 \times 7\ H_2O$, 41.5 mg/L KI, 12.1 mg/L $Na_2MoO_4 \times 2\ H_2O$, 1.25 mg/L $CoSO_4 \times 5\ H_2O$, 1.46 mg/L $Co(NO_3)_2 \times 6\ H_2O$). For cultivation on solid medium, 12 g/L Agar was added to the KnopME medium. Moss suspension cultures were disrupted weekly with an ULTRA-TURRAX (IKA) at 18,000 rpm for 90 s. If not indicated otherwise, moss was grown under standard light conditions (55 µmol photons/m²s) at 22 °C in a 16 h/8 h light/dark cycle.

Hydroponic Physcomitrella gametophore cultures were assembled as described[36,127]. Here, a thin layer of protonema from suspension was distributed on gauze mesh (PP, 250 m mesh, 215 m thread, Zitt Thoma GmbH, Freiburg, Germany) capped on a glass ring. The glass rings with protonema-covered mesh gauze were placed in Magenta®Vessels (Sigma-Aldrich, St. Louis, USA) and KnopME medium was added until the protonema-covered gauze mesh was moist. The medium was exchanged every 4 weeks. Gametophores were harvested after 12 weeks.

Gametophore colonies on Agar plates (KnopME) were generated by transplanting single gametophores to new plates. Plates were sealed with Parafilm®.

**Generation of AARE knockout lines**. Knockout constructs for each *PpAARE* gene were generated by amplifying genomic regions as homologous flanks. The PCR products were fused to a neomycin (*PpAARE1*), hygromycin (*PpAARE2*), and zeocin (*PpAARE3*) resistance cassettes, respectively, employing triple template PCR as described[128] with primer sequences listed in Supplementary Data S1 using the Q5 polymerase (New England Biolabs, Ipswich, USA). The knockout constructs were released from their vector backbones with XhoI (*PpAARE1*), BglII (*PpAARE2*) and DraI (*PpAARE3*), respectively. Enzymes were purchased from Thermo Fisher Scientific. Digested plasmids were precipitated and sterilized prior to transfection using standard ethanol precipitation method[129]. Transfection of Physcomitrella WT or KO (for consecutive knockouts) protoplasts was conducted via PEG-mediated procedure[77,130]. The WT strain as well as the identified *PpAARE* KO lines are accessible via the International Moss Stock Center (IMSC, www.moss-stock-center.org). IMSC accession numbers for the mutants and WT are available in Supplementary Data S4.

Screening of plants surviving antibiotics selection was done by PCR. KO mutant plants surviving the antibiotics selection were checked for the presence of a genomic region which should be removed upon homologous recombination-based integration of the knockout construct (Fig. S4a). In case of the absence of this WT signal, plants were further checked for correct 5′ and 3′ integration of the respective knockout construct using primers listed in Supplementary Data S1.

**Protonema growth and gametophore induction**. Suspension cultures were started at the same day and disrupted weekly as described. Dry weight was measured in triplicates and suspension density was adjusted to 440 mg dry weight (DW) per litre (mg DW/L) as described[131]. Droplets of 15 µL were distributed on solid medium (Knop ME, 0.02% glucose). Sealed plates were cultivated as described above. Three droplets each per line were distributed on one plate and all lines were grown on the same plate. 4 plates (12 colonies) were used per assay. Colony areas were measured with *ImageJ*. White pixels counted from binarized colony images were used as quantitative values. Gametophores were counted upon visibility of the first leafy structure on buds.

**Generation of PpAARE-Citrine knock-in lines**. Knock-in constructs to fuse the coding sequence of Citrine to the C-terminus of PpAARE isoforms via an 18 bp linker[128] at the endogenous genomic locus were cloned via Gibson assembly[132]. All necessary parts were amplified using primers listed in Supplementary Data S1. Additionally, XhoI (*PpAARE1*), SalI (*PpAARE2*), and BamHI (*PpAARE3*) restriction sites were added to the 5′ and 3′ ends of the respective knock-in construct. All parts were assembled into pJet1.2 vector backbone (Thermo Scientific) using the Gibson Assembly®Cloning Kit from New England Biolabs (Ipswich, Massachusetts, USA). Transfection of Physcomitrella WT protoplasts were conducted as described[77,130] by co-transfecting a plasmid containing a neomycin phosphotransferase resistance (nptII) cassette as transient selection marker (pBSNNNEV, Mueller et al.)[57]. The linearized plasmid and the co-transfection vector were purified and sterilized via ethanol precipitation[129] prior to transfection.

The presence of Citrine was checked with primers listed in Supplementary Data S1 and resulting positive lines were further checked for correct 5′ and 3′ integration by PCR using the Plant Phire Kit with primers listed in Supplementary Data S1. All identified fusion lines are available via the International Moss Stock Center (IMSC, www.moss-stock-center.org). IMSC accessions are listed in Supplementary Data S4.

**qPCR analysis**. The copy number of the integrated KO constructs was determined using a qPCR-based method[78]. Genomic DNA was extracted from 100 mg frozen protonema using the GeneJET Plant Genomic DNA Purification Kit (Thermo Scientific, Waltham, USA). DNA concentrations were adjusted to 3 ng/µL for each sample and qPCR was performed with primers for the 5′ and 3′ flanks as well as with primers for the corresponding selection cassette. Additionally, primers for the single copy gene *CLF* (Pp3c22_2940V3) were used as internal control. Reference lines were WT as well as in-house lines with known single integrations of the used selection cassettes. Primers are listed in Supplementary Data S5. PCR reactions were done using 2x SensiFAST Mix (Bioline, London, UK) and analyzed in a Lightcycler 480 II (Roche, Basel, Schweiz).

**cDNA preparation**. RNA was extracted using the innuPREP Plant RNA Kit (Analytik Jena, Jena, Germany). The extracted RNA was treated with DNAse I (Thermo Scientific) and subsequently reverse transcribed into cDNA using Superscript III Reverse Transcriptase (Life Technologies, Carlsbad, USA).

**Fusion constructs for subcellular localization**. All constructs were generated using Gibson assembly[132] and integrated into a PpAct5:Linker:eGFP-MAV4 vector backbone[73]. Integration of the different coding sequences was done in frame in front of the Linker:eGFP. All parts for the Gibson assembly (inserts and corresponding vector backbones) were amplified either with Phusion™ polymerase (Thermo Fisher Scientific) or with HiFi polymerase (PCR Biosystems Ltd) according to the manufacturer's instructions. The primers were designed to have a 25 bp overlap to the corresponding fragment to be fused with. All primers and combinations are listed in Supplementary Data S6. In the case of the N-terminal

difference of AtAARE ($M^1$-$A^{55}$ of AT4G14570.1, gene model provided by TAIR (https://www.arabidopsis.org/)) the Actin5 promoter was replaced by the CaMV35S promoter[133] previously validated in Physcomitrella[134].

Cloned plasmids were purified using the PureYield™ Plasmid Midiprep kit (Promega, Wisconsin, USA) according to the manufacturer´s instructions. The plasmids were purified and sterilized via ethanol precipitation[129].

**Confocal microscopy**. Confocal imaging was performed on transiently transfected live protoplasts using Leica TCS SP8 (Leica Microsystems, Wetzlar, Germany). Immediately before microscopy, MitoTracker™ Orange CMTMRos (Thermo Fisher Scientific) was added to protoplast suspensions to a final concentration of 100 nM. For all imaging experiments, an HC PL APO CS2 63x/1.40 OIL objective was used with a zoom factor of 4. The pinhole was set to 35.4 µm. For excitation, a WLL laser (70%) was used. In a sequential acquisition setup, eGFP and chlorophyll were excited with the same laser beam (488 nm, 2%) and their signals were detected simultaneously, whereas MitoTracker™ was excited with a different laser beam (550 nm, 2%) and its signal was detected separately. The detection ranges were specified as 502–546 nm for eGFP, 662–732 nm for chlorophyll, and 597–636 nm for MitoTracker™. The images were acquired as z-stacks with the number of optical sections varying with the protoplast size. The voxel sizes of the resulting z-stacks were 0.0903, 0.0903, 0.239 µm in the x-y-z order. For visual representation and analysis, single slices with the best signal-to-noise ratio were selected and extracted from each z-stack using FIJI software.

**Cultivation of Arabidopsis**. Seeds were surface-sterilized for 4 min in 80% ethanol and subsequently for 1 min in 100% ethanol. Seeds were placed on plates containing ½ MS supplemented with 1% (D + ) sucrose and 0.8% Agar. Alternatively, seeds were directly placed on sterilized soil. Seeds were incubated at 8 °C for 2–4 days in the dark for stratification before placing them in growth chambers. Plants were further cultivated under short day conditions at 22 °C and 70 µmol photons/m²s in an 8 h/16 h light/dark cycle. For bolting assays, plants were placed in a phytochamber at 22 °C and 70 µmol photons/m²s in a 16 h/8 h light/dark cycle (long day condition). Rosette leaves and days since sowing were counted upon appearance of the shoot in the middle of the rosette.

**Screening of Arabidopsis AARE mutants**. *Arabidopsis thaliana* lines with T-DNA insertions in the At4G14570 locus were identified from the public T-DNA Express database at the SIGnAL website (Salk Institute Genomic Analysis Laboratory). Lines GK-456A10, SALK_080653C, SALK_071125C, and SALK_205137C, were obtained from the Nottingham Arabidopsis Stock Centre. Homozygous mutant alleles were verified by PCR using the following primers: forward LB GK 5′-ATATTGACCATCATACTCATTGC-3′ and reverse GK-456 5′-CTTCAAAGAAACACCAATCAG-3′ for the GK-456A10 line, and forward LB-pROK 5′-GCGTGGACCGCTTGCTGCAACT-3′ and reverse Salk_53 5′-TCTTTAGCCGAATCAGTTCCAGA-3′ for the SALK_080653C, SALK_071125C, and SALK_205137C lines. Identified homozygous mutant plants were back-crossed with Arabidopsis Col-0. The F2 generation was screened for homozygous mutant plants using the above listed primer sets to identify the mutant allele or substituting the forward primers with forward WT-GK-456 5′-AAGATGCTTTGCAGTCTC TA-3′ and forward WT-Salk 5′-ACTGCCTTATGATCCATTGTCTC-3′, to identify the GK and SALK lines WT alleles, respectively. RT-PCR was additionally performed to check for the presence of *AtAARE* transcripts using Taq polymerase on cDNA prepared as described above with primers At1-At4. All primer combinations are listed in Supplementary Data S4.

**AARE exopeptidase activity**. The enzyme activity assay according to[17] was modified. Here, tissue (80–100 mg) was homogenized in liquid nitrogen and dissolved in 1 mL extraction buffer (50 mM PBS, 1 mM EDTA, 2 mM DTT). After centrifugation at 20,000 × *g* for 20 min at 4 °C, 300 µL supernatant was mixed with 700 µL reaction buffer (50 mM HEPES-KOH, pH 7.5 containing 1 mM AcAla-pNA (Bachem, Bubendorf, Switzerland) or 50 mM HEPES-KOH, pH 7.5, 10% DMSO containing 1 mM AcLeu-pNA (Bachem) and incubated at 37 °C for 30–120 min. The reaction was stopped by the addition of 500 µL 30% acetic acid. Absorbance was measured at 410 nm in a photospectrometer. This approach was used to generate the data of Figs. 3b and S6a–d. Later the method was modified (Figs. 3g and S6e-g). Here, tissue was homogenized in liquid nitrogen and dissolved in 100 µL extraction buffer (50 mM PBS, 1 mM EDTA, 2 mM DTT) per 10 mg FW. After centrifugation at 20,000×*g* for 20 min at 4 °C, 5 µL supernatant was mixed with 195 µL reaction buffer (50 mM HEPES-KOH, pH 7.5 containing 1 mM AcAla-pNA (Bachem, Bubendorf, Switzerland)) in a 96 well micro titer plate and incubated at 37 °C for 30–120 min. In the case of AcLeu-pNA as substrate, 50 mM HEPES-KOH, pH 7.5 with 10% DMSO containing 1 mM AcLeu-pNA (Bachem) was used as reaction buffer. Every biological sample was measured in three technical replicates. Absorbance was measured at 410 nm. Activity was calculated using a molar absorbance coefficient[21] of 8.8 mM*cm$^{-1}$ and represents the release of pNA [µmol] per minute normalized to the total protein concentration of the sample. The protein concentration was determined using the $A_{280}$ method of a NanoDrop™ (Thermo Fisher Scientific) or with a Bradford assay[135].

**Western blots**. Western blots were performed as described[74] using the ECL Advance detection kit (GE Healthcare). The primary antibody against *A. thaliana* AARE was kindly provided by Dr. Yasuo Yamauchi[17]. The primary antibody was diluted 1:10,000 in TBST Buffer with 2% Blocking (GE Healthcare) and incubated on the membrane for 2 h. As secondary antibody anti-Guinea pig antibody, coupled to a horseradish peroxidase (Agrisera, AS 10 1496) diluted 1:10,000 in 2% TBST with 2% Blocking (GE Healthcare), was applied for 1 h.

**Detection of oxidized proteins**. Plant tissues were homogenized in liquid nitrogen and proteins were extracted in 50 mM PBS, 50 mM DTT, 1 mM EDTA. In all, 3–6 µg total protein was derivatized with DNPH and subsequently detected with an anti-DNP antibody according to the manufacturer's instruction of the OxyBlot Protein Oxidation Detection Kit (S7150, Sigma-Aldrich). Equal amounts of the derivatized protein samples were employed as loading control on a separate SDS-gel and stained with Coomassie or silver staining.

**Flow cytometry**. Flow cytometry analysis was performed as described[79]. Here, protonemata were chopped with a razor blade in a small petri dish (6 cm diameter) in 2 mL of DAPI-buffer containing 0.01 mg/L 4′,6-Diamidin-2-phenylindol (DAPI), 1.07 g/L MgCl₂ × 6 H₂O, 5 g/L NaCl, 21.11 g/L Tris, 0.1% Triton, pH 7. The solution was filtered using 30 µm pore size filters and the fluorescence intensity was measured using a Cyflow®Space flow cytometry system (Partec, Munich, Germany).

**Computational predictions**. Predictions for the presence of cleavable targeting peptides were performed with *TargetP2.0*[63]. Additional predictions of subcellular localizations were performed with *LOCALIZER*[64]. The presence of peroxisomal targeting signals was predicted with *PredPlantPTS1*[66,67]. Prediction of protein domains was performed using *InterProScan*[136] and protein domain annotations according to PFAM[137] were used.

**Co-Immunoprecipitation**. Co-immunoprecipitation was performed using GFP-Trap Magnetic Particles M-270 (Chromotek, Planegg-Martinsried, Germany) as recommended by the manufacturer with modifications. In all, 300 mg protonema was homogenized in a 2 mL reaction tube using a tungsten and a glass bead. For each line three biological replicates were realized. The extraction buffer was chosen according to the manufacturer's recommendations for plant samples and contained 25 mM HEPES-KOH, pH 7.5, 2 mM EDTA, 100 mM NaCl, 200 nM DTT, 0.5% Triton X-100, 1% plant protease inhibitor cocktail (PPI, P9599, Sigma Aldrich). Ground plant material was dissolved in a final volume of 2 mL ice-cold extraction buffer and incubated for 10 min in a water quench ultrasonic device. Samples were centrifuged at 4 °C at 20,000 × *g* for 30 min. For each sample 25 µL magnetic particle slurry was washed with 500 µL extraction buffer. The sample supernatant was transferred to the cleaned beads and incubated, rotating for 1 h at 6 °C. Subsequently, beads were washed with 1 mL extraction buffer without Triton and PPI and again with 500 µL. Beads were then dissolved in 500 µL wash buffer (10 mM Tris-HCl, pH 7.5, 150 mM NaCl, 0.5 mM EDTA), transferred to a new reaction tube and washed again in 500 µL wash buffer. A RapiGest solution (0.2% in 50 mM Tris-HCl, pH 7.5; RapiGest SF Surfactant (Waters, Milford, MA, USA) was mixed 3:1 with 5 mM DTT in 50 mM Tris-HCl, pH 7.5. 30 µL of the resulting mixture was applied to each sample. Samples were incubated at 95 °C for 5 min under continuous shaking. Samples were cooled down to RT and 5 µL of a trypsin (V5117, Promega) solution (0.025 µg/µL in 50 mM Tris-HCl, pH 7.5) were added to each sample. Digestion on the beads was performed for 30 min at 32 °C under continuous shaking. Supernatants were transferred to new reaction tubes and the remaining beads were washed twice with 50 µL 5 mM Iodoacetamide solution (in 50 mM Tris-HCl, pH 7.5). The wash supernatants were combined with the trypsin-containing supernatant and incubated over night at 30 °C under continuous shaking. Acid-catalyzed cleavage of the RapiGest surfactant was performed as recommended by the manufacturer. Samples were purified using C18-STAGE-Tips as described[138] and eluted from the tip in 30% ACN in 0.1% FA.

**Mass spectrometry measurement and data analysis**. MS analysis was performed on an Orbitrap Q-Exactive Plus instrument (Thermo Fisher Scientific) coupled to an UltiMate 3000 RSLCnano system (Dionex LC Packings/Thermo Fisher Scientific) as described[139]. Database search and label-free quantitation was performed using *MaxQuant* software V1.6.0.16[140]. For each Co-IP a specific database was employed containing all V3.3 proteins of Physcomitrella[39] as well as the sequence of the respective fusion protein. Additionally, the contaminant list provided by the software was included. Decoys were generated on the fly by reverting the given protein sequences. Variable modifications were formation of pyro Glu (N-term Q, −17.026549 Da), oxidation (M, + 15.994915 Da), acetylation (N-term, + 42.010565 Da) and deamidation (N, + 0.984016 Da). Carbamido-methylation (C, + 57.021464 Da) was specified as fixed modification. Enzymatic specificity was set to tryptic with semi-specific free N-terminus. An FDR of 0.01 was set for protein identification. LFQ values[141] were used as quantitative values. Interaction analysis was performed in Perseus V1.6.12.0[142]. Missing values were imputed from a normal distribution with a down-shift of 1.8 and distribution

width of 0.3. Interaction partners were accepted at an FDR of 0.01 and a p-value < 0.01.

Raw files of the test-IP measurements (Fig. S8e) were processed using Mascot Distiller V2.7.10 and searched against all Physcomitrella protein models V3.3[39] using Mascot Server V2.7.0 (Matrix Science). Processed mgf files from immunoprecipitation experiments targeting N-terminal arginylation (PXD003232[36,143]) and the test-IPs were searched again against all Physcomitrella protein models V3.3[39] using Mascot Server V2.7.0 (Matrix Science). The precursor mass tolerance was 5 ppm and the fragment mass tolerance was 0.02 Da. Variable modifications were formation of pyro Glu (N-term Q, −17.026549 Da), oxidation (M, + 15.994915 Da), acetylation (N-term, + 42.010565 Da) and deamidation (N, + 0.984016 Da). Carbamidomethylation (C, + 57.021464 Da) was specified as fixed modification. Enzymatic specificity was set to tryptic with semi-specific free N-terminus. Search results were loaded in Scaffold[TM] 5 (V5.0.1, Proteome Software) and proteins were accepted at an FDR = 1 and peptides at an FDR = 0.5. A table of identified proteins is accessible in Supplementary Data S7.

**Multiple sequence alignment and phylogenetic reconstruction**. Homologous protein sequences were aligned with UPP[144] (version 4.4.0) using default parameters and subsequently translated into a codon-aware CDS alignment with PAL2NAL[145] (version 1.4). Based on this multiple coding sequence alignment we reconstructed a maximum likelihood tree with RAxML[146] (version 8.2.12) using the GTRCAT model with 1000 rapid bootstrap samples. The tree was rooted at the split between animal and plant sequences and plotted in R[147] using the packages ape[148] and ggtree[149].

**Statistics and reproducibility**. Statistical differences in datasets were analyzed with one-way Anova with subsequent post-hoc test if different lines at same conditions were compared. Two-way Anova with subsequent post-hoc test was performed to analyze differences between lines at different conditions. Sample sizes of biological replicates are specified in the figure legends. Anova and post-hoc analysis was performed in R[147]. Statistical significance was accepted at $*p < 0.05$, $**p < 0.01$, and $***p < 0.001$.

**Reporting summary**. Further information on research design is available in the Nature Portfolio Reporting Summary linked to this article.

## Data availability
The authors confirm that all relevant data supporting the findings of this study are available within the paper and its supplementary files. The mass spectrometry proteomics data have been deposited to the ProteomeXchange Consortium via the PRIDE partner repository[84,85] with the dataset identifier PXD033854 and 10.6019/PXD038742. Plant lines used in this study are available upon reasonable request from the corresponding author (R.R.) or *via* the International Moss Stock Center (IMSC, www.moss-stock-center.org). IMSC accessions are listed in Supplementary Data S4. Full Blot images for Fig. 6b and Fig. S3b are available in Fig. S10. Supplementary Data S8 contains all numeric source data used to generate the graphs and charts in this study. Plasmids generated in this study are available from the International Moss Stock Center IMSC (https://www.moss-stock-center.org) with the accession numbers P1519 (PpAARE1 KO (pJet)), P1841 (PpAARE2 KO (pJet)), P1520 (PpAARE3 KO (pJet)), P1655 (PpAARE1:Citrine KI (pJet)), P1813 (PpAARE2:Citrine KI (pJet)), P1814 (PpAARE3:Citrine KI (pJet)), P1833 (PpAARE1_1:eGFP (pMAV4)), P1834 (PpAARE1_2:eGFP (pMAV4)), P1853 (PpAARE1_Nt:eGFP (pMAV4)), P1855 (PpAARE3:eGFP (pMAV4)), P1856 (PpAARE3:eGFP (pMAV4)), P1854 (AtAARE_SV:eGFP (pMAV4)), P1881 (AtAARE_LV:eGFP (pMAV4)) and P1862 (AtAARE_Nt:eGFP (pMAV4)).

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

## Acknowledgements

We thank Christine Glockner, Agnes Novakovic and Eija Schulze for expert technical assistance and Anne Katrin Prowse for language editing. Support with the Arabidopsis assays from Dr. Philipp Schwenk is gratefully acknowledged. We thank Prof. Dr. Bettina Warscheid for the possibility to use the QExactive Plus instrument and Prof. Dr. Yasuo Yamauchi for the primary antibody against *A. thaliana* AARE. We gratefully acknowledge funding by the Deutsche Forschungsgemeinschaft (DFG, German Research

Foundation) under Germany's Excellence Strategy EXC-2189 (CIBSS to R.R.) and by the Wissenschaftliche Gesellschaft Freiburg.

## Author contributions

S.N.W.H. designed research, performed experiments, analyzed data, and wrote the manuscript. B.Ö. and N.v.G. analyzed data and helped writing the manuscript. A.A.M. analyzed data. B.R.v.B., L.N., J.S.F., R.K., S.G., T.W., and F.S. performed experiments. M.R.F. and S.J.M.S. designed research and helped writing the manuscript. R.R. designed and supervised research, acquired funding, and wrote the manuscript. All authors approved the final version of the manuscript.

## Funding

## Competing interests

The authors declare no competing interests.
