## [Peer Review File · Communications Biology]

Reviewers' comments:

Reviewer #1 (Remarks to the Author):

The manuscript entitled "A deeply conserved protease, acylamino acid-releasing enzyme (AARE), acts in plant ageing" aims at describing a common, universal aging mechanism by extending our knowledge on protein oxidation in animals (including humans) into plants. It appears to me, however, that this work has a major experimental flaw and this is the selection of species used. In my opinion, one cannot evaluate plant aging and talk about a "unified concept of aging" (as stated in the abstract and throughout) by examining a moss (*Physcomitrella*) and an annual, short-lived angiosperm (*Arabidopsis*). Trying to establish a unified concept of aging requires the selection and investigation of other plant groups, including gymnosperms and very importantly plant species with contrasting maximum life spans. Otherwise it seems to me that conclusions cannot be supported by data. Alternatively, in case additional experiments are not possible, I strongly suggest authors fully redefine the aims of the study and entirely rewrite the manuscript.

Reviewer #2 (Remarks to the Author):

Hoernstein et al. present a comprehensive analysis of AAREs in plants. They identified three isoforms in *Physcomitrella*, which form hetero-oligomeric complexes in vivo. One of these isoforms, PpAARE1, is targeted to chloroplasts, mitochondria and the cytosol, as the result of an alternative splicing event. The single *Arabidopsis* isoform is also localized to these three compartments, mediated by alternative initiation of translation. The authors further show that PpAARE1 contributes most to AARE exopeptidase activity, while endopeptidase activity is accounted for by PpAARE2. Accelerated aging is observed in AARE loss-of-function mutants in both *Physcomitrella* and *Arabidopsis*. PpAARE1,2 double, and 1,2,3 triple mutants show faster progression from protonema to gametophores and reduced life span of gametophores, which correlated with the accumulation of oxidized proteins. Enhanced levels of oxidized proteins were also observed in AtAARE mutants, and increased protein oxidation appeared to be linked to a faster transition to flowering. The authors conclude that the function of AARE in protein homeostasis and aging is conserved between plants and animals.

This is a well conceived and carefully conducted study. The findings are novel and highly interesting. The conclusions are generally well supported by the data, often including multiple independent lines of evidence. The paper is very well written and will be of considerable interest to a broad audience, not limited to plant biologists. I only have a few minor comments that may help to further improve the manuscript.

1. In my view, the proposition that protein oxidation may act as a signal for aging, particularly for high temperature-induced changes in gene expression, is a bit of a stretch. Correlations are observed, but a cause-effect-relationship between protein oxidation and aging-related phenotypes has not been fully established. Therefore, I suggest to tone this down (e.g. last sentence of the Results section) and to delete the penultimate paragraph of the discussion.
2. The manuscript is rather long and may benefit from shortening. E.g. in the Results section, where more than two pages are devoted to the phylogenetic analysis. Some rather trivial explanations are also not needed (e.g. on page 21 '...bolting, which is the swift upward growth at the transition to flowering.' In the Discussion, the last sentence on page 22 ('Chloroplasts are a major ...') and the second but last paragraph can be deleted.
3. Nalpha-acetylation is known to have a major impact on proteome stability and stress responses (doi:10.1126/sciadv.abn6153, doi:10.1038/s41467-022-28414-5). This should be discussed with respect to the AAREs role in protein turnover and exopeptidase activity.
4. On page 3, '...derivatives with ROS-generated lipid peroxides' – is this what is meant: '...adducts with ROS-generated reactive aldehydes'?

5. Please define acronyms on first use (e.g. CDS, NLS, FPKM)
6. On page 13, you refer to Fig S4b when you mention the qPCR results; panels j-p should also be included.
7. On page 13: AARE exopeptidase activity is not proven, but assayed via
8. On page 14: `... as they develop gametophores from protonema earlier (Fig. 3d)'. Time to gametophore development is not shown in Fig. 3d.
9. On page 21: `together, we found AARE exopeptidase and endopeptidase activity in our experiments'. Should this rather read ` Together, we found exopeptidase and endopeptidase activity to be reduced in AtAARE mutants.'?
10. Figs. 3b,g; 6c; S6: include the unit of measure at the y-axis
11. Fig. S4 j-p: is this correct? Some lines contain more than 100 or even 1000 integrations?

Reviewer #3 (Remarks to the Author):

The manuscript of S. Hoernstein et al. describes a very thorough study of the three AARE genes in the moss *Physcomitrella* and, for comparison, of the single gene in the angiosperm *Arabidopsis*. The analysis of the function and subcellular localization of the corresponding proteins is described by well-designed experiments, as well as the approach for understanding their relative interactions. The spearhead of this work is certainly the generation of mutants with loss of function, which were used here for the first time in the study of this protease family, which allowed to find correlations between AARE function, accumulation of oxidized proteins and signs of aging.

Here are some observations that can contribute to the discussion of the results, plus a few suggestions for spelling improvements.

1) Results session, Page 5

"Stress conditions result in a decreased AARE expression in *Physcomitrella* and *Arabidopsis* (Fig. S1c-e)".

I do not agree with this sentence because for the PpAARE genes the time considered for the effects on their expression may be too short (1 hour, 37 ° C), while for AtAARE gene we can see a modulation in the expression which also includes upregulation phases, as for the expression patterns of *apehSs* and *apeh-3Ss* genes in response to stressful conditions described in Gogliettino et al., 2012.

2) Results session, Page 7

"In addition, this analysis reveals a closer relationship between PpAARE1 and PpAARE3, which presumably originate from a more recent gene duplication event, compared to PpAARE2. This is supported by the fact that the open reading frames (ORFs) of PpAARE1 and PpAARE3 are represented by a single exon whereas the ORF of PpAARE2 is split across 17 exons, similar to AtAARE (Fig. 1b)." With respect to the intron-rich PpAARE2, PpAARE1 and PpAARE3 may have a more recent origin, as described for intronless genes in "Liu H. et al. The emergence and evolution of intron-poor and intronless genes in intron-rich plant gene families. *Plant J.* 2021 Feb;105(4):1072-1082", that seem to play more likely a role in response to stress or participate in epigenetic processes and plant development. What is described in this work for PpAARE1 goes in the same direction.

3) Results session, Page 5

"PpAARE isoforms are in paralogs" should be:
PpAARE isoforms are in paralogs.

4) Methods session, Page 28

"...the Actin5 promotor was replaced by the CaMV35S promotor..." should be:
...the Actin5 promoter was replaced by the CaMV35S promoter...

Reviewer #1 (Remarks to the Author):The manuscript entitled "A deeply conserved protease, acylamino acid-releasing enzyme (AARE), acts in plant ageing" aims at describing a common, universal aging mechanism by extending our knowledge on protein oxidation in animals (including humans) into plants. It appears to me, however, that this work has a major experimental flaw and this is the selection of species used. In my opinion, one cannot evaluate plant aging and talk about a "unified concept of aging" (as stated in the abstract and throughout) by examining a moss (*Physcomitrella*) and an annual, short-lived angiosperm (*Arabidopsis*). Trying to establish an unified concept of aging requires the selection and investigation of other plant groups, including gymnosperms and very importantly plant species with contrasting maximum life spans. Otherwise it seems to me that conclusions cannot be supported by data. Alternatively, in case additional experiments are not possible, I strongly suggest authors fully redefine the aims of the study and entirely rewrite the manuscript.

Response:

It is not our intention **to present a unified concept of ageing** in a single paper. Instead, we show that AARE function in a moss and an angiosperm is correlated with ageing. As the evolutionary distance between mosses and seed plants is more than 500 million years, this is a deep evolutionary conservation. Moreover, our findings correlate well with findings about oxidized proteins, AARE, and stress / ageing in a variety of animals. Therefore, we hope **to contribute to a unified concept of ageing** with our paper. We have made this clearer now in our revised version.

Reviewer #2 (Remarks to the Author):

Hoernstein et al. present a comprehensive analysis of AAREs in plants. They identified three isoforms in *Physcomitrella*, which form hetero-oligomeric complexes in vivo. One of these isoforms, PpAARE1, is targeted to chloroplasts, mitochondria and the cytosol, as the result of an alternative splicing event. The single *Arabidopsis* isoform is also localized to these three compartments, mediated by alternative initiation of translation. The authors further show that PpAARE1 contributes most to AARE exopeptidase activity, while endopeptidase activity is accounted for by PpAARE2. Accelerated aging is observed in AARE loss-of-function mutants in both *Physcomitrella* and *Arabidopsis*. PpAARE1,2 double, and 1,2,3 triple mutants show faster progression from protonema to gametophores and reduced life span of gametophores, which correlated with the accumulation of oxidized proteins. Enhanced levels of oxidized proteins were also observed in AtAARE mutants, and increased protein oxidation appeared to be linked to a faster transition to flowering. The authors conclude that the function of AARE in protein homeostasis and aging is conserved between plants and animals.

This is a well conceived and carefully conducted study. The findings are novel and highly interesting. The conclusions are generally well supported by the data, often including multiple independent lines of evidence. The paper is very well written and will be of considerable interest to a broad audience, not limited to plant biologists. I only have a few minor comments that may help to further improve the manuscript.

1. In my view, the proposition that protein oxidation may act as a signal for aging, particularly for high temperature-induced changes in gene expression, is a bit of a stretch. Correlations are observed, but a cause-effect-relationship between protein oxidation and aging-related phenotypes has not been fully established. Therefore, I suggest to tone this down (e.g. last sentence of the Results section) and to delete the penultimate paragraph of the discussion.

Response:

We changed accordingly. Line 568-569: The last sentence of the results section was shortened to “Consequently, this accelerated developmental transition in the mutants correlates with enhanced levels of oxidized proteins.”

Line 686-697: We deleted the penultimate section of the discussion.

2. The manuscript is rather long and may benefit from shortening. E.g. in the Results section, where more than two pages are devoted to the phylogenetic analysis. Some rather trivial explanations are also not needed (e.g. on page 21 ‘...bolting, which is the swift upward growth at the transition to flowering.’ In the Discussion, the last sentence on page 22 (‘Chloroplasts are a major ...’) and the second but last paragraph can be deleted.

Response:

We changed accordingly and shortened the first results section where possible. Further, line 551-552 was changed to “To corroborate this, we investigated levels of oxidized proteins in the *Arabidopsis* T-DNA mutant (s68) in comparison to WT (Col) cultivated under short day conditions. “

The following sentence “We analysed plants cultivated under short day conditions to delay bolting, which is the swift upward growth at the transition to flowering.” was deleted.

3. Nalpha-acetylation is known to have a major impact on proteome stability and stress responses (doi:10.1126/sciadv.abn6153, doi:10.1038/s41467-022-28414-5). This should be discussed with respect to the AAREs role in protein turnover and exopeptidase activity.

Response:

The AARE exopeptidase function on intact proteins apart from short peptides was not clearly proven, although it is indicated in Tsunasawa et al. 1975 (<https://doi.org/10.1093/oxfordjournals.jbchem.a130722>). However, Adibekian et al. 2011 (<https://doi.org/10.1038/nchembio.579>) show an alteration of the N-terminal modification state but no effect on protein stability after AARE inhibition. This is another hint towards exopeptidase function on intact proteins, but it is not in line with a potential role on protein stability. Therefore, we want to perform proteomic investigations of our AARE mutants in future to investigate a potential role of AARE as modulator of protein stability by acting on N^{alpha} acetylation of proteins. We did not include this in the discussion in our manuscript since we do not show any data that could contribute to answering this question. To our knowledge the global role of N^{alpha} acetylation of proteins remains controversial and an introduction into this background would unnecessarily expand our manuscript. Therefore, we prefer not to include this discussion in our manuscript.

4. On page 3, ‘...derivatives with ROS-generated lipid peroxides’ – is this what is meant: ‘...adducts with ROS-generated reactive aldehydes’?

Response:

Line 66 was changed to: “ROS lead to irreversible cysteine oxidation, advanced glycation end-products, adducts with ROS-generated reactive aldehydes’, ...”

5. Please define acronyms on first use (e.g. CDS, NLS, FPKM)

Response:

Changed accordingly.

6. On page 13, you refer to Fig S4b when you mention the qPCR results; panels j-p should also be included.

Response:

The reference to Fig. S4b is meant to show the scheme depicting the primer pairs used for the qPCR. Panels j-p are mentioned two sentences later referring specifically to the results of the qPCR experiments.

7. On page 13: AARE exopeptidase activity is not proven, but assayed via

Response:

Changed accordingly.

Line 364 was changed to “Typically, AARE exopeptidase activity is assayed *via* N^{alpha}-acetylated amino acids like AcAla...”

8. On page 14: ‘... as they develop gametophores from protonema earlier (Fig. 3d)’. Time to gametophore development is not shown in Fig. 3d.

Response:

In this experiment (Fig. 3d), we cannot exactly say when the first gametophores were visible. The image was taken 9 days after distributing protonema suspension culture on plates as specified in the figure legend. At this time, we observed that the double KO of PpAARE1/2 as well as the triple KO already showed gametophores, whereas the WT and other KO lines did not. Consequently, we subsequently performed the quantitative comparison and checked the presence of gametophores at the indicated timepoints (Fig. 3f) and thus present the data you asked for in this experiment.

9. On page 21: ‘together, we found AARE exopeptidase and endopeptidase activity in our experiments’. Should this rather read ‘ Together, we found exopeptidase and endopeptidase activity to be reduced in AtAARE mutants.’?

Response:

Line 560 was changed to “Together, we found AtAARE exopeptidase and endopeptidase activity to be reduced in our experiments.”

10. Figs. 3b,g; 6c; S6: include the unit of measure at the y-axis

Response:

Changed accordingly.

Fig. 3:

Fig. 6:

Fig. S6:

11. Fig. S4 j-p: is this correct? Some lines contain more than 100 or even 1000 integrations?

Response:

Some transformed moss lines contain more than 100 and up to 1000 integrations of the KO-construct. This is a known but unwanted side effect of the transfection procedure. Since the chance of off-target effects in lines with such a high number of integrations is high, we performed this analysis and subsequently analyzed only lines with a single integration. We explain this now in the revised version (line 360-362).

Reviewer #3 (Remarks to the Author):

The manuscript of S. Hoernstein et al. describes a very thorough study of the three AARE genes in the moss *Physcomitrella* and, for comparison, of the single gene in the angiosperm *Arabidopsis*. The analysis of the function and subcellular localization of the corresponding proteins is described by well-designed experiments, as well as the approach for understanding their relative interactions. The spearhead of this work is certainly the generation of mutants with loss of function, which were used here for the first time in the study of this protease family, which allowed to find correlations between AARE function, accumulation of oxidized proteins and signs of aging.

Here are some observations that can contribute to the discussion of the results, plus a few suggestions for spelling improvements.

1) Results session, Page 5

“Stress conditions result in a decreased AARE expression in *Physcomitrella* and *Arabidopsis* (Fig. S1c-e)”.

I do not agree with this sentence because for the PpAARE genes the time considered for the effects on their expression may be too short (1 hour, 37 ° C), while for AtAARE gene we can see a modulation in the expression which also includes upregulation phases, as for the expression patterns of *apehSs* and *apeh-3Ss* genes in response to stressful conditions described in Gogliettino et al., 2012.

Response:

We agree, the time of the applied stress treatments in *Physcomitrella* might be too short to state “Stress conditions result in a decreased AARE expression...”. Additional stress treatments might be necessary to support this statement. Nevertheless, the stress treatment in *Arabidopsis* (38°C) lasted 3 hours but the timepoints in the graph specify sampling under recovering conditions at 25°C (Kilian et al., 2007, <https://doi.org/10.1111/j.1365-313X.2007.03052.x>). Therefore, the expression pattern indicates down-regulation in response to the heat treatment rather than a modulation, while a recovery after 6 hours at 25°C can be observed. In order not to expand the manuscript unnecessarily we did not go too much into detail at this point, since we also do not include any stress-related experiments. We now specify this more precisely in the figure legend of the supplement and modified line 133-135 to:

“Stress conditions decrease AARE expression in *Arabidopsis* shoots and in *Physcomitrella* protonemata.”

2) Results session, Page 7 “In addition, this analysis reveals a closer relationship between PpAARE1 and PpAARE3, which presumably originate from a more recent gene duplication event, compared to PpAARE2. This is supported by the fact that the open reading frames (ORFs) of PpAARE1 and PpAARE3 are represented by a single exon whereas the ORF of PpAARE2 is split across 17 exons, similar to AtAARE (Fig. 1b).” With respect to the intron-rich PpAARE2, PpAARE1 and PpAARE3 may have a more recent origin, as described for intronless genes in “Liu H. et al. The emergence and evolution of intron-poor and intronless genes in intron-rich plant gene families. *Plant J.* 2021 Feb;105(4):1072-1082”, that seem to play more likely a role in response to stress or participate in epigenetic processes and plant development. What is described in this work for PpAARE1 goes in the same direction.

Response:

We are grateful for this valuable input, we added a reference to the work of Liu *et al.* in line 185-186 and added “This is in congruence with a more recent emergence of intron-poor genes in intron-rich families linked to stress response and developmental processes...”. Further, this is in line with our previous work on intron-less orphan moss genes as earliest responders to abiotic stress (Beike *et al.* 2014, DOI 10.1111/nph.13004). We have added this reference too (line 187).

3) Results session, Page 5 “PpAARE isoforms are in paralogs” should be: PpAARE isoforms are in paralogs.

Response:

We changed the sentence in line 136-137 to “To investigate whether other plants also possess multiple AARE homologs and to infer their phylogenetic relation, ...”.

4) Methods session, Page 28“...the Actin5 promotor was replaced by the CaMV35S promotor...” should be:...the Actin5 promoter was replaced by the CaMV35S promoter...

Response:

Changed accordingly.

Reviewers' comments:

Reviewer #1 (Remarks to the Author):

It would be nice to consider reviewers' opinions when revising a manuscript. Authors refer to "plant ageing" in the title and state that results "thus contribute to a unified concept of ageing" at the end of the abstract, while stating in your rebuttal letter that your work did not intend so, while recognizing that using a moss and an annual plant is not enough to talk generally about plant ageing. The work will clearly confound readers and it appears to me that authors have no intention to avoid it.

Reviewer #2 (Remarks to the Author):

The authors addressed the points that were raised by the reviewers and revised their manuscript accordingly. I find the manuscript much improved and have no further suggestions.

Reviewer #3 (Remarks to the Author):

The authors responded adequately to all my comments. The manuscript can be published in its latest version.

Reviewer #1 (Remarks to the Author):

It would be nice to consider reviewers' opinions when revising a manuscript. Authors refer to "plant ageing" in the title and state that results "thus contribute to a unified concept of ageing" at the end of the abstract, while stating in your rebuttal letter that your work did not intend so, while recognizing that using a moss and an annual plant is not enough to talk generally about plant ageing. The work will clearly confound readers and it appears to me that authors have no intention to avoid it.

Changed title to: "A deeply conserved protease, acylamino acid-releasing enzyme (AARE), acts in ageing in Physcomitrella and Arabidopsis".

Rephrased to: "suggest a unified concept of ageing may exist in different life forms".

Added as last paragraph of the discussion: "To solidify the role of AARE in ageing in different life forms, particularly in plants with contrasting maximum life spans and in animals of different complexity, loss-of-function mutants should be established and analysed in selected model species. Likewise, a deeper understanding of AARE-function in human diseases is desirable. Together, such analyses may contribute to a unified concept of ageing in different life forms."

Reviewer #2 (Remarks to the Author):

The authors addressed the points that were raised by the reviewers and revised their manuscript accordingly. I find the manuscript much improved and have no further suggestions.

Thank you very much.

Reviewer #3 (Remarks to the Author):

The authors responded adequately to all my comments. The manuscript can be published in its latest version.

Thank you very much.

Final Revision Instructions

To the Author— Please review the editorial comments and requests below and confirm that changes have been made in the manuscript in the right-hand column. **This document must be uploaded as a related manuscript file.**

Please see our final file submission checklist for information about submitting your revised documents.

Files and General Policies	
Main manuscript file must be in Microsoft Word or LaTeX format. LaTeX and Tex article source files must be accompanied by the compiled PDF for reference. The bibliography must be submitted separately (as a .bib file) or contained within the .tex file.	Confirmed.
Each Figure must be provided as a separate file and must be supplied whole, with all panels included in a single document. Figures should be provided at a minimum resolution of 300 dpi at final size. Figure files must only contain images (please also leave out labels such as “Figure 1” etc). Figure captions must instead be included within the main manuscript file, grouped together at the end of the document.	Confirmed.
All figures, tables, and supplementary items must be cited in the manuscript and numbered in the order in which they appear.	Confirmed.
Please check whether your manuscript contains third-party images, such as figures from the literature, stock photos, clip art or commercial satellite and map data. We strongly discourage the use or adaptation of previously published images, but if this is unavoidable, please request the necessary	The manuscript does not contain third-party images.

rights documentation to re-use such material from the relevant copyright holders and return this to us when you submit your revised manuscript. An appropriate permissions statement must be present in the relative figure caption for any third-party images.	
Please check that you have not copied any text directly from published work (even your own) without clear attribution, including one or more references. We run a plagiarism detection software and may need to request additional changes if we identify large blocks of identical text.	Confirmed.
An updated editorial policy checklist that verifies compliance with all required editorial policies must be completed and uploaded with the revised manuscript. All points on the policy checklist must be addressed; if needed, please revise your manuscript in response to these points. https://www.nature.com/documents/nr-editorial-policy-checklist.pdf. Please note that this form is a dynamic 'smart pdf' and must therefore be downloaded and completed in Adobe Reader. This file will not open in an internet browser.	Done.
The reporting summary will be published alongside your manuscript therefore it needs to accurately represent your work. In this case, please take a closer look at the reporting summary and make sure things are completed correctly. If an item does not apply, for example human participants, I need you to check the NA box next to that item. No section should be left blank. Also, please make sure to include your name and date at the top of the document. If you require a new Reporting Summary form, please download it here: https://www.nature.com/documents/nr-reporting-summary.pdf.	Done.

Please note that this form is a dynamic 'smart pdf' and must therefore be downloaded and completed in Adobe Reader. This file will not open in an internet browser.	
Your paper will be accompanied by a brief editor's summary when it is published on our homepage. Please approve the draft summary below or provide us with a suitably edited version (no more than 250 characters including spaces). The analysis of the function of acylamino acid-releasing enzymes (AARE) in Physcomitrella and Arabidopsis reveals a connection between AARE and plant ageing, suggesting a unified concept of ageing may exist across domains of life.	Approved.
ORCID Communications Biology is committed to improving transparency in authorship. As part of our efforts in this direction, we are now requesting that all authors identified as 'corresponding author' create and link their Open Researcher and Contributor Identifier (ORCID) with their account on the Manuscript Tracking System (MTS) prior to acceptance. ORCID helps the scientific community achieve unambiguous attribution of all scholarly contributions. For more information please visit http://www.springernature.com/orcid. For all corresponding authors listed on the manuscript, please follow the instructions in the link below to link your ORCID to your account on our MTS before submitting the final version of the manuscript. If you do not yet have an ORCID you will be able to create one in minutes.	Done.

https://www.springernature.com/gp/researchers/orcid/orcid-for-nature-research IMPORTANT: All authors identified as ‘corresponding author’ on the manuscript must follow these instructions. Non-corresponding authors do not have to link their ORCIDs but are encouraged to do so. Please note that it will not be possible to add/modify ORCIDs at proof. Thus, if they wish to have their ORCID added to the paper they must also follow the above procedure prior to acceptance. To support ORCID's aims, we only allow a single ORCID identifier to be attached to one account. If you have any issues attaching an ORCID identifier to your MTS account, please contact the Platform Support Helpdesk at http://platformsupport.nature.com/	
We regularly highlight papers published in Communications Biology on the journal’s Twitter account (@CommsBio). If you would like us to mention authors, institutions, or lab groups in these tweets, please provide the relevant twitter handles in the right-hand column.	@ReskiLab
We would welcome the submission of material for the ‘Featured Image’ section on the Communications Biology home page. Images should relate to the content of your manuscript but need not be contained within the paper. Photographs and aesthetically interesting images are preferred; diagrams are generally not used. Suggestions should be uploaded as a Related Manuscript file. Please provide 1200x675-pixel RGB images. You will also need to submit a completed Image License to Publish. Unfortunately, we cannot promise that your suggestions will be used.	Provided.
Supplementary information	

Supplementary Information Format and referencing  ● Supplementary Figures, small Tables, and any supplementary text must be provided in a single PDF. Figures and their captions should be presented together.  ○ If you include a title page, please check that the title and author list matches the main manuscript. ● All Supplementary items must be referred to in the manuscript, and items must be mentioned in numerical order. Please do not include general references to “Supplementary Material”; instead refer to specific items. ● Additional files can be provided as Supplementary Data (Excel files, text files, .zip folders), Supplementary Movies, Supplementary Audio, or Supplementary Software (.zip folder) Supplementary Information files will be uploaded with the published article as they are submitted with the final version of your manuscript. Any highlighting or tracked changes should be removed from the file.	Done.
Supplementary items must be cited in a consistent format. Names of items in the Supplementary file(s) must match those used in the main manuscript. We recommend using the following naming formats: Supplementary Figure 1, Supplementary Table 1, Supplementary Data 1, Supplementary Note 1, and Supplementary References.	Done.

Large tables and other data types: We strongly recommend depositing these to suitable repositories (such as Figshare, Dryad, or a data type-specific repository if one exists). Otherwise, these must be supplied as Supplementary Data files. Each file must be labelled as Supplementary Data 1, etc. Please either move large supplementary tables (e.g. Table S1, S2 and S7) or all supplementary tables to Supplementary Data. Please modify the citations accordingly.	All tables were moved to Supplementary Tables (S1-S7), labelled and cited accordingly.
It's mandatory to provide access to the numerical source data (raw data) for graphs and charts: We strongly recommend depositing these to suitable repositories (such as Figshare, Dryad, or a data type-specific repository if one exists). Otherwise, all source data underlying the graphs and charts presented in the main figures must be uploaded as Supplementary Data (in Excel or text format). Note that only the data used directly for generating the charts needs to be supplied. Please provide the source data for Fig. 3b, 3e-3g, 5b, 6c, 6e, S1, S3c, S4j-S4p, S5, S6, S8d, S8e, S8g as Supplementary Data.	All numerical source data are now provided in Supplementary Table S8. We specify this in the data availability statement.
For any Supplementary Files such as those mentioned above that are not included your combined PDF (e.g. Supplementary Data, Movies, Audio, Software), please provide a title and description for each file here in the column to the right. For example: File name: Supplementary Data 1 Description: The source data behind the graphs in the paper	Supplementary Table S1 List of primers used to assemble KO and KI constructs and primers used to screen for transgenic plants in Physcomitrella. Supplementary Table S2 List of gene accession numbers for all sequences used for

communications biology

phylogenetic analysis and corresponding subcellular localization predictions.

Supplementary Table S3 List of primers used to screen possible T-DNA mutant plants in Arabidopsis.

Supplementary Table S4 List of accession numbers for the International Moss Stock Center (IMSC) of all transgenic Physcomitrella lines.

Supplementary Table S5 List of primers used for qPCR on genomic DNA to determine copy numbers of KO constructs in transgenic Physcomitrella lines.

Supplementary Table S6 List of primers used to assemble the constructs for transient localization analysis in Physcomitrella protoplasts.

Supplementary Table S7 Spectrum report of the revised database search of the anti-Arg^(Nt)

	IP published in Hoernstein et al., 2016. (doi: 10.1074/mcp.M115.057190) Supplementary Table S8 This table contains all numeric source data used to generate the graphs in Fig. 3b, Fig. 3e-g, Fig. 5b, Fig. 6c, Fig. 6e, Fig. S1, Fig. S3c, Fig. S4j-p, Fig. S5, Fig. S6, Fig. S8d, Fig. S8e and Fig. S8g.
Supplementary References should appear at the end of the Supplementary Information file. Numbering must start from 1. If a supplementary reference also appears in the main manuscript reference list, please repeat it in the Supplementary References.	Confirmed.
Title Page	
Please ensure that the author list provided in our manuscript tracking system matches the author list in the main manuscript.	Confirmed.

Please modify your author list and affiliations to comply with the following: ● Author tagging statements are limited to the following two options: "These authors contributed equally" and "These authors jointly supervised this work", with no more than one of each tag permitted.● “Present address” must be provided separately as final affiliations. For example, Buğra Özdemir^{1*} should be Buğra Özdemir^{1,5}. 5 will be the present address.	Done.
Manuscript title Please ensure the title clearly describes the central finding of the paper. We recommend writing the title as a declarative statement of approximately 15 words or fewer. Be sure to include any key species, protein names, or gene names to ensure optimal retrieval of the paper in database searches.	Confirmed.
Main text	

Format of the main text

Please ensure your manuscript includes the following sections, presented in this order:

1. **“Introduction”**: The background and rationale for the work. The final paragraph should be a brief summary of the major results and conclusions. The results of the current study must only be discussed in this final paragraph. The Introduction should contain no references to figures or tables. Do not include subheadings.
2. **“Results” or “Results and Discussion”**. This should be split into subheaded sections; we recommend 1 subheading per main figure or table. Figures should not be embedded in the text but submitted separately.
 - a. Do not use more than 1 layer of subheadings.
 - b. A “Conclusions” paragraph can be included **only if the results and discussion are combined into a single section.**
3. **“Discussion” (optional)**, without subheadings.
4. **Methods**, which should be split into subheaded sections. Do not use more than 1 layer of subheadings.

To improve readability, we recommend that the main text (Introduction, Results and Discussion) be limited to approximately 5000 words or fewer.

Please move figure legends from the Results section to the end of manuscript and remove underlines in the Methods section.

Figure legends were moved to the end of the manuscript and underlines in the method section were removed.

Statistical reporting

Wherever statistics have been derived (e.g. error bars, box plots, statistical significance) **the legend needs to provide and define the n number** (i.e. the sample size used to derive statistics) as a precise value (not a range), using the wording “n=X biologically independent samples/animals/independent experiments” etc. as applicable.

Please provide and define n numbers in Fig. 3f, S1a, S1c-S1e.

N numbers for each box in **Fig. 3f** have been placed directly in the plot.

N-numbers in **Fig. 1a** differ, this is specified now in the legend and indicated by the addition of the individual datapoints. Error bars are only plotted for n=3.

N-numbers for **Fig. S1c** have been specified in the figure legend.

Fig. S1d and **1e** are based on biological duplicates according to the public eFP browser

(http://bar.utoronto.ca/efp_arabidopsis/).

The single replicate values are not available.

The error bars have been removed and the number of replicates is specified in the figure legend.

Statistical representation

Statistics such as error bars cannot be derived from $n < 3$ and must be removed from all such cases. We strongly discourage deriving statistics from technical replicates, and they should be removed from all such cases, unless there is a clear scientific justification for why providing this information is important. Conflating technical and biological variability, e.g. by pooling technically replicate samples across independent experiments is strongly discouraged.

Please remove error bars in Fig. S6f, S8e if they are from $n=2$.

Error bars in **Fig. S6f** were removed as well as the significance levels.

Error bars in **Fig. 8e** were removed, individual datapoints were added.

Please check that all gene and mRNA names are in italics. Protein names should not be in italics. Please confirm that only official gene/protein symbols are used and that species names are in italics.	Confirmed. Arabidopsis and Physcomitrella are common names of Arabidopsis thaliana and Physcomitrella patens, which was renamed to Physcomitrium patens recently. Common names are in upright letters, botanical species names in italics.
Display items	
Figure captions/legends Figures must have a title that will appear above the Figure and a legend that will appear below the Figure (see e.g. https://www.nature.com/articles/s42003-020-1059-1/figures/1) The Figure title must describe the Figure as a whole and must not contain reference to specific figure panels. The Figure legend must refer to and describe all panels. Abbreviations, symbols, colors, and shading present in the Figure must be defined. Please write out the symbols/colors in words (blue circles, red dashed line, etc.) within these definitions. All figure panels must be labelled using lower case letters. Please refrain from referring to sections of figures as top/bottom/left/right/, etc. Please remove brackets around figure panel labels. Please remove “(b).” after “(a, d, e, and g).” in Fig. S6 legend.	All brackets around figure panel labels have been removed.
Axis and panel labels will be published as received. We recommend using a sans-serif font such as Arial or Helvetica.	Accordingly.

Data presentation in bar graphs and line graphs

For all graphs depicting a single point value (e.g., mean) with error bars, **you must add individual data points or convert the graph to a boxplot or dot-plot**. You may wish to refer to this blog post about representing data distribution in plots (particularly for small datasets). We strongly encourage the same for plots with multiple time courses depicted. See the June 24, 2019 CommsBio editorial for more details about this policy. Example plots are shown here:

Examples of plots showing data distribution. Figure 2 from the editorial linked to above.

Please individual data points for Fig. S1a, S1c-S1e, S3c, S8d, S8e.

Individual data points obtained from PEATmoss (<https://peatmoss.plantcode.cup.uni-freiburg.de/>) were added for **Fig. S1a** and **Fig S1c**.

For **Fig. S1b** we added the individual data points available from Athena database (<http://athena.proteomics.wzw.tum.de/>) which were used to calculate the plotted means.

Fig. S1d and **1e** are based on biological duplicates according to the public eFP browser (http://bar.utoronto.ca/efp_arabidopsis/).

The single replicate values are not available. The error bars have been removed and the number of replicates is specified in the figure legend.

In **Fig. 3c** individual data points were added. The bar chart in **Fig. 8d** represents single values which are only depicted to show that all isoforms have been identified in those IP experiments. They are not used to derive a quantitative statement. In **Fig. 8e** values from biological duplicates are shown, we removed the error bars and added the individual data points.

Please define the error bars in each Figure and Supplementary Figure where they are used. One statement at the end of each Figure caption is sufficient if the error bars are equivalent throughout the Figure. Please define boxplots in Fig. 3e, 3f, 6e.	The boxplots were defined accordingly in the figure legends.
Microscopy images and photographs in each Figure and Supplementary Figure must be accompanied by scale bars, and these must be defined.	Confirmed.

Blots and gels All blots/gels must be accompanied by size markers in every figure panel. Uncropped and unedited blot/gel images must be included as Supplementary Figure(s). The new Supplementary Figure(s) must be cited in the main manuscript text (for example, in the Data Availability Statement). Please pay close attention to our Digital Image Integrity Guidelines and to the following points below:  ● that unprocessed scans are clearly labelled and match the gels and western blots presented in figures. Unprocessed scans must be included in a supplementary figure. ● that control panels for gels and western blots are appropriately described as loading on sample processing controls ● all images in the paper are checked for duplication of panels and for splicing of gel lanes. Finally, please ensure that you retain unprocessed data and metadata files after publication, ideally archiving data in perpetuity, as these may be requested during the peer review and production process or after publication if any issues arise. Please provide uncropped blot images for Fig. 6b and S3b as Supplementary Figure 10. Please cite it in the Data Availability statement.	Uncropped images of Fig. 6b and Fig. S3b are provided in Fig. S10 and cited in the data availability statement.
Methods	
Please ensure that all information present in the Reporting Summary is also in the manuscript. This information is usually most appropriate in the Methods section.	Confirmed.

We allow unlimited space for Methods. The Methods must contain sufficient detail such that the work could be repeated. It is preferable that all key methods be included in the main manuscript, rather than in the Supplementary Information. Please avoid use of “as described previously” or similar, and instead detail the specific methods used with appropriate attribution.	Accordingly.
The Methods should include a separate section titled “Statistics and Reproducibility” with general information on how the statistical analyses of the data were conducted, and general information on the reproducibility of experiments (also those lacking statistical analysis), including the sample sizes and number of replicates and how replicates were defined.	Confirmed.
Data Policies	
The Data Availability statement must include:  ● Access details for deposited data, including repository name and unique data ID. ● How source data can be obtained. ● A statement that all other data are available from the corresponding author (or other sources, as applicable) on reasonable request. Note that ‘available upon request’ is only appropriate if immediate data access has not been mandated by our policies or by the editors. See here for more information about formatting your Data Availability Statement: http://www.springernature.com/gp/authors/research-data-policy/data-availability-statements/12330880	Confirmed.

Mandatory deposition of raw and processed data is required for:

- All sequencing data (DNA, RNA, protein)
- Novel human genetic polymorphisms (e.g., dbSNP)
- Linked genotype and phenotype data (e.g., dbGaP for human data)
- GWAS summary statistics or polygenic risk scores
- Novel macromolecular structure
- Gene expression microarray data (must be MIAME compliant)
- Crystallographic data for small molecules
- Mass spectrometry-based proteomics data

For more information on mandatory data deposition policies at the Nature Portfolio, please visit

<http://www.nature.com/authors/policies/availability.html#data>

For an up-to-date list of approved repositories for each mandatory data type, please visit <https://www.springernature.com/gp/authors/research-data-policy/repositories/12327124>.

Accession code(s) for deposited data must be provided in the Data Availability statement in the final version of the paper. Failure to do so will delay publication. Please ensure data are available prior to publication.

Accordingly.

Communications Biology has a strong preference for all data to be deposited in an approved repository. In some cases, data deposition may be required by the editor. We recommend the following data repositories:  ● GenBank (all DNA sequence data) ● NHGRI-EBI GWAS Catalog (GWAS summary statistics) ● PGS Catalog (polygenic risk scores) ● Gene Expression Omnibus (Microarray or RNA sequencing data) ● Sequence Read Archive (WGS or WES data) ● Protein Data Bank (protein structural data) ● OSF (neuroimaging raw data and EEG/EMG/MEG raw data) ● Neurovault (unthresholded statistical maps, parcellations, and atlases produced by MRI and PET studies) ● Image Data Resource (microscopy data) ● PRIDE (proteomics data) Data types without a specific repository can be deposited in a generalist repository, such as figshare or Dryad. For an up-to-date list of approved repositories, please visit https://www.springernature.com/gp/authors/research-data-policy/repositories/12327124.	Accordingly.
Data citation Please cite datasets stored in external repositories in the main reference list. For previously published datasets, we ask authors to cite both the related research articles and the datasets themselves.	Accordingly.

For more information on how to cite datasets in submitted manuscripts, please see our data availability statements and data citations policy.	
Please deposit your newly generated plasmids in a community repository (eg, Addgene). Include the ID numbers in the Data Availability statement.	Accordingly.
End Notes	
Please check that your bibliography complies with the following:  ● Your bibliography should start with the heading “References”. The references must be numbered in the order of appearance in the text, then tables, then figures. ● Any in-text citations to references (e.g. "Gupta et al. show...") should be followed by their corresponding reference citation number from the reference list. ● Manuscript citations must include journal title, article title, volume number, page or article number or DOI, and year of publication. ● No publication can be present more than once in the reference list. ● No footnotes are permitted in the references or elsewhere. Text should be incorporated into the main text, the Methods section, or the Supplementary Information instead. ● Websites should only be listed in the references if they are in common use or curated. ● Where possible, preprints in the reference list should be updated with details of the published, peer-reviewed paper. ● Citations should be formatted in the text using superscript numbers. 	Accordingly.

Please provide a 'Competing interests' statement using one of the following standard sentences: • The authors declare the following competing interests: [specify competing interests]• The authors declare no competing interests. See our competing interests policy for further information: https://www.nature.com/nature-research/editorial-policies/competing-interests Please use standard sentence.	Confirmed.
Please check that your 'Author Contributions' section individually lists the specific contribution of each author to the work. Each author must be referred to by name or initials. Where multiple authors possess identical initials, they must be clearly disambiguated from one another. See our author contributions policy for further information: https://www.nature.com/nature-research/editorial-policies/authorship#author-contribution-statements	Accordingly.
No separate funding section is permitted. Please include your funding information in Acknowledgements instead.	Confirmed.